



# Examining the impacts of estimated precipitation isotope ($\delta^{18}$O) inputs on distributed tracer-aided hydrological modelling

Carly J. Delavau[1], Tricia Stadnyk [1], Tegan Holmes[1]

[1]Department of Civil Engineering, University of Manitoba, Winnipeg, R3T 5V6, Canada.

Correspondence to: Carly J. Delavau (Carly.Delavau@gov.mb.ca)

**Abstract.** Tracer-aided hydrological models are becoming increasingly popular tools as they have documented utility in constraining model parameter space during calibration, reducing model uncertainty, and assisting with selection of appropriate model structures. However, the issue of data availability, particularly input data, proves to be a major challenge associated with this type of application. Tracer-aided hydrological modelling typically requires a time series of isotopes in precipitation ($\delta^{18}$O$_{ppt}$) to drive model simulations, but unfortunately, throughout much of the world, and particularly in sparsely populated high-latitude regions, these data are not widely available. This study uses the isoWATFLOOD tracer-aided hydrological model to investigate the usefulness of three types of estimated $\delta^{18}$O$_{ppt}$ for model input, and the impact that these data have on model simulations and parameterization in the remote Fort Simpson Basin, NWT, Canada. This study showed that although total simulated streamflow was not significantly impacted by choice of $\delta^{18}$O$_{ppt}$ input, isotopes in streamflow ($\delta^{18}$O$_{SF}$) simulations and the internal apportionment of water (and therefore, model parameterizations) were impacted, particularly during large precipitation and snowmelt events. This finding highlighted the importance of estimated $\delta^{18}$O$_{ppt}$ to capture both the variability and seasonality in precipitation isotopes as critical for tracer-aided hydrological modelling, especially when precipitation events displayed distinctly different isotopic compositions than that of streamflow. This study achieves an understanding of how isoWATFLOOD can be used in regions with a limited number of $\delta^{18}$O$_{ppt}$ observations, and that the model can be of value in such regions. This study reinforces that a tracer-aided modelling approach assists with resolving hydrograph component contributions, and works towards diagnosing the issue of model equifinality.

## 1 Introduction

Hydrological models are critical tools for the planning, development, design, operation and sustainable management of water resources (Singh and Frevert, 2006). These models provide insight into applications such as the prediction of floods, droughts and water availability, and the effects of climate and land use change on water resources. Problems arise in calibrating and validating hydrological model predictions, primarily due to: (1) a lack of available data at sufficient resolutions to force and validate model simulations - especially in remote, high-latitude locations (in Canada: Coulibaly et al., 2013); (2) model parameterization and issues with equifinality; and (3) the associated uncertainty in model results (Beven and Binley, 1992; Kirchner, 2006; Fenicia et al., 2008; Dunn et al., 2008).





It is now widely accepted that calibration and validation of hydrological models based solely on streamflow is not a sufficient evaluation measure (Kuczera, 1983; Beven and Binley, 1992; Kuczera and Mroczkowski, 1998; Seibert and McDonnell, 2002; Kirchner, 2006; Fenicia et al., 2008; Dunn et al., 2008). Modellers are focusing on a model's ability to

correctly partition, store and release water from hydrologic compartments, in addition to adequately simulating total streamflow response. Conservative tracer data provides insights into the dominant hydrological processes and integrated runoff response (in northern catchments: Birks and Gibson, 2009; Tezlaff et al., 2015), and such data assist with constraining model parameter space during calibration, reducing model uncertainty, and assisting with selection of appropriate model structures (e.g., Tetzlaff et al., 2008; Birkel et al., 2010a; McMillian et al., 2012; Birkel et al., 2014; Smith et al., 2016). An

increasing number of studies have investigated the utility of tracer-aided modelling approaches, especially over the past decade (for a comprehensive overview, see Birkel and Soulsby, 2015).

Although greatly informative, previous tracer-aided modelling studies have generally been conducted using lumped conceptual rainfall-runoff models in highly instrumented small-scale experimental catchments ($<10^2$ km$^2$). This has resulted

in distributed studies at the regional-scale ($>10^3$ km$^2$) left largely unexplored, with the exception of a few, select applications (Stadnyk et al., 2013). Modelling at the regional-scale typically requires a distributed approach to capture the heterogeneity in meteorological inputs, basin characteristics, and runoff response, resulting in more complex, highly parameterized models (e.g., Michaud and Sorooshian, 1994; Carpenter and Georgakakos, 2006; Her and Chaubey, 2015). Because it is at these larger scales where models are applied operationally and management decisions are based, there is a critical need to

understand the abilities, limitations, and uncertainties associated with distributed tracer-aided modelling at the regional scale.

Although there is an identified need, the issue of data availability, particularly input data, proves to be a major challenge associated with this type of application (Birkel and Soulsby, 2015). Tracer-aided hydrological modelling typically requires a time series of isotopes in precipitation ($\delta^{18}O_{ppt}$) to drive model simulations. Unfortunately, throughout much of the world,

and particularly in sparsely populated high-latitude regions (such as the vast majority of Canada), these data are not widely available. Although automatic samplers are becoming increasingly common, watersheds in which snow accumulation is substantial will continue to be fraught with difficulties surrounding the collection and characterization of precipitation isotopes, particularly during the winter months (Dietermann and Weiler, 2013; Penna et al., 2014). The lack of spatial and temporal density of $\delta^{18}O_{ppt}$ observations highlights the need for alternative methods to provide estimates of stable isotopes in

precipitation for tracer-aided model forcing. Options include empirically-based models generating gridded time series estimates of precipitation isotopes (e.g., Lykoudis et al., 2010; Delavau et al., 2015), in addition to isotope-enabled climate model output (for a comprehensive overview, see: Noone and Sturm, 2010; Xi et al., 2014).



Small-scale catchment studies rely on continuous records of $\delta^{18}O_{ppt}$ observations at high temporal frequencies (typically daily, and less commonly, weekly) for model input. At the larger scale, tracer-aided modelling completed by Stadnyk et al. (2013) in the remote Fort Simpson region of northern Canada used annual average compositions of rainfall and snowfall $\delta^{18}O$ to drive model simulations. Their results suggested that utilizing annual, spatially static oxygen-18 in precipitation forcing has the potential to significantly impact simulations and consequently, model parameterization as well. The assumption that model forcing is spatially invariant is not preferable, as $\delta^{18}O_{ppt}$ can vary drastically over small space and time scales due to changes in moisture sources and transport processes, rainout history and seasonality (e.g., in Canada: Gat et al., 1994; Moran et al., 2007; Birks and Edwards, 2009).

Utilizing estimates of $\delta^{18}O_{ppt}$ to drive tracer-aided model simulations has not yet been investigated, particularly at the regional-scale where the lack of observed data have identified a need for estimates of $\delta^{18}O_{ppt}$. Therefore, this study aims to explore how estimated $\delta^{18}O_{ppt}$ inputs of differing spatial and temporal variability impact regional tracer-aided modelling within two remote Canadian basins, where limited $\delta^{18}O_{ppt}$ observations exist. Our specific research objectives are to:

a) force a regional tracer-aided hydrological model (isoWATFLOOD) with three types of estimated $\delta^{18}O_{ppt}$ input of varying temporal and spatial resolution;

b) examine how choice of $\delta^{18}O_{ppt}$ input affects the simulation of total streamflow and isotopes in streamflow ($\delta^{18}O_{SF}$), while accounting for parameter uncertainty, and;

c) explore if choice of $\delta^{18}O_{ppt}$ input affects the internal apportionment of water, namely the seasonality of hydrograph components.

## 2 Study area and data

### 2.1 The Fort Simpson Basin

The Fort Simpson Basin (FSB) is located within the Lower Liard River valley close to the town of Fort Simpson, Northwest Territories, Canada (61°45 N; 121°14 W; Fig. 1). This region has been the focus of several tracer-aided hydrological studies (e.g., St Amour et al., 2005; Stadnyk et al., 2005; 2013; Stadnyk-Falcone, 2008). The FSB is selected for this study to build upon previous modelling work conducted within the region, and follow up on recommendations from Stadnyk et al. (2013) suggesting further analysis and improvement of isoWATFLOOD $\delta^{18}O_{ppt}$ input. The study period of 1997–1999 is selected based data availability.

This study considers two sub-basins of the greater Fort Simpson basin: the Jean-Marie and Blackstone River sub-basins (Fig. 1). Differences in wetland distribution and function, basin physiography and land cover make-up between the two watersheds (Table 1) are the primary reasons in selecting these sub-basins for this study. These marked differences ensure





that watersheds of varying dominant hydrological processes are represented in the modelling, and therefore the impacts of $\delta^{18}O_{ppt}$ input selection on these processes can be examined.

The land cover classification breakdown (Table 1) shows the primary land cover type within the sub-basins as transitional, consisting of shrubs, deciduous varieties and early generation spruce. The region has a high proportion of wetlands, with the total wetland percentage in Table 1 representing both bogs (disconnected drainage) and fens (connected drainage); although the amount of each type within each respective sub-basin varies. Aylsworth and Kettles (2000) state that Jean-Marie is predominately fen peatlands, while Blackstone is bog-dominated peatlands, with very few or no fen peatlands present.

The Ecoregions Working Group (1989) classifies the FSB as a sub-humid mid- to high-boreal ecoclimatic region (Hbs), classified by cool summers approximately five months in length, with moderate (300-500 mm) annual precipitation. Winters are very cold with persistent snow cover. The hydrological response is dominated by snowmelt during late April to early May, while summer and fall runoff events are due to major rainfall, with a return to baseflow occurring during dry summer periods or towards the beginning of the ice-on season in October.

## 2.2 Meteorological and hydrometric data

Daily total precipitation, mean daily temperature, and hourly relative humidity data are obtained from Environment Canada's Fort Simpson Airport weather station. Observed precipitation is supplemented with ANUSPLIN-derived daily precipitation extracted at eight locations throughout the Fort Simpson region (Fig. 1). ANUSPLIN is a multidimensional non-parametric surface fitting method that has been found well suited to the interpolation of various climate variables, particularity in data-sparse, high-elevation regions as the method accounts for spatially varying dependencies on elevation (McKenney et al., 2011). An inverse-distance weighting approach is used spatially distribute the daily ANUSPLIN and observed precipitation time series across the model domain (Kouwen, 2014). Rainfall that occurred over the study period, particularly in 1997, was significantly higher than normal. Additionally, 1998 was above average in temperature, which is especially prevalent in the first portion of the year. Other researchers have attributed the increased rainfall and warmer temperatures to a strong El Niño influence from mid-1997 to mid-1998 (Petrone et al., 2000; St Amour et al., 2005).

Hydrometric records are obtained from Water Survey of Canada. Jean Marie was gauged at Highway No.1 in 1972 with a period of record of 44 years, whereas Blackstone was gauged at Highway No.7 in 1991 having a record length of 25 years. Neither sub-basin is regulated, therefore all flows are considered to be natural. During the study period, mean annual discharge was above normal in both sub-basins in 1997, normal in Jean Marie and slightly below normal in Blackstone in 1998, and below normal in both sub-basins in 1999. A statistical summary of observations used in this study is provided in Table 4.





## 2.3 Isotope data

During 1997 to 1999, intensive sampling took place in the Fort Simpson Basin as part of the Mackenzie Study of the Global Energy and Water Experiment (GEWEX; Stewart et al., 1998). The campaign sampled $\delta^{18}$O and $\delta^2$H of streamflow, rainfall, snowpack, and surface waters (wetlands and lakes) during the open water season (May to October). During ice-on conditions, the isotope stratigraphy of river ice extracted during late March in 1998 and 1999 was used to reconstruct the isotopic composition of winter streamflow (Gibson and Prowse, 1999; Prowse et al., 2002; St Amour et al., 2005). This study uses measured $\delta^{18}$O compositions in streamflow ($\delta^{18}$O$_{SF}$) in the Jean-Marie (n = 71) and Blackstone (n = 69) sub-basins for model calibration. Although $\delta^{18}$O$_{ppt}$ compositions (n = 27) were collected as part of the GEWEX sampling campaign, their spatial and temporal resolutions are not adequate for model forcing. These data are incorporated into the study as a means to validate simulated $\delta^{18}$O$_{ppt}$ forcing input, when available. The number of measurements and their statistical properties are summarized in Table 4. Isotopic compositions of $\delta^{18}$O are expressed in delta ($\delta$) notation as a deviation from VSMOW (Vienna Mean Standard Mean Ocean Water) in units of per mille (‰), such that such that $\delta_{water} = (R_{water}/R_{VSMOW} - 1) \times 1000$ ‰, where R is $^{18}$O/$^{16}$O in the sample and standard, respectively. Isotope samples were analyzed at the Environmental Isotope Laboratory at the University of Waterloo, and St Amour et al. (2005) indicated maximum analytical uncertainties of $\pm$ 0.1 ‰ for $\delta^{18}$O.

## 2.4 Precipitation oxygen-18 input

The default method for $\delta^{18}$O$_{ppt}$ input in isoWATFLOOD is annual average compositions of rainfall and snowfall $\delta^{18}$O for each year of the simulation. Values for the FSB are obtained through averaged measurements of $\delta^{18}$O in rainfall and snowpack from the GEWEX study (Table 2; Table 4). $\delta^{18}$O$_{ppt}$ compositions are assumed constant throughout the watershed domain. Due to the averaged values and lack of spatial variability, this input type is referred to as *static* throughout the remainder of the manuscript.

Times series simulations obtained from the KPN43 model created by Delavau et al. (2015) are used as a secondary $\delta^{18}$O$_{ppt}$ input. The KP43 model uses North American Regional Reanalysis (NARR; Mesinger et al., 2006) climate variables, teleconnection indices, and geographic information to produce gridded time series of oxygen-18 in precipitation at a monthly time step. This input is generated at a 10 km resolution (to mirror model set-up), and varies spatially throughout the study area due to the variation in the climatic predictors and geographic information required to produce simulations.

The third $\delta^{18}$O$_{ppt}$ input assessed in this study is regional climate model output from the isotope-enabled climate model, REMOiso (Sturm et al., 2005; Sturm et al., 2007). Raw REMOiso $\delta^{18}$O$_{ppt}$ output is available at a 55 km spatial resolution and a 6h time step. However, REMOiso output is averaged to a daily time step, as the range and variability of sub-daily $\delta^{18}$O$_{ppt}$



are erroneously large, and the resolution of streamflow oxygen-18 calibration data do not warrant a temporal frequency of input finer than daily.

### 2.4.1 REMOiso bias correction

Due to a lack of published studies evaluating REMOiso performance within Canada, a comparison between REMOiso output and Canadian Network for Isotopes in Precipitation observations (CNIP; Birks and Gibson, 2009) is completed to determine if REMOiso simulations require a regional bias correction. CNIP data are now part of the Global Network for Isotopes in Precipitation (GNIP) database and can be accessed at: http://www.iaea.org/water (IAEA, 2014). This analysis is completed at Snare Rapids, NWT, the closest CNIP station to the FSB, for the years of 2000 and 2001. Snare Rapids is located approximately 330 km northeast of Fort Simpson and has monthly $\delta^{18}O_{ppt}$ observations spanning the years of 1997–2010. A longer time frame of comparison between CNIP and REMOiso is not possible due to the short overlapping period of REMOiso simulations and CNIP observations. Daily REMOiso simulations are averaged to monthly compositions for direct comparison to CNIP data using the precipitation amount-weighting approach in Eq. (1):

$$\delta^{18}O_{ppt\ monthly} = \sum P_i * (\delta^{18}O_{ppt})_i / \sum P_i \tag{1}$$

where $P_i$ is the amount of daily precipitation (mm) obtained from the Snare Rapids Canadian Air and Precipitation Monitoring Network (CAPMoN) station operated by Environment Canada, where isotopic compositions are also sampled.

Uncorrected REMOiso simulations exhibit a positive bias in this region (Fig. 2), which is expected based on the ECHAM4 mean annual $\delta^{18}O_{ppt}$ output (Noone and Sturm, 2010) and personal communications with S. J. Birks and K. Sturm (2016). Therefore, a seasonal bias correction is applied to daily REMOiso simulations. The bias correction is calculated as the average seasonal difference between the monthly amount-weighted REMOiso output and the CNIP observations (Table 3). Corrected monthly and daily REMOiso output at Snare Rapids are displayed on Fig. 2 as the dashed red and solid orange lines, respectively. For the current study, daily REMOiso output for the Fort Simpson region is bias corrected with the seasonal values in Table 3. The statistical properties of the corrected daily REMOiso simulations, alongside the KPN43 monthly simulations and the static seasonal averages are summarized in Table 4.

### 3 Methods

### 3.1 Background and set-up

The tracer-aided hydrological model used in this study is isoWATFLOOD (Stadnyk-Falcone, 2008; Stadnyk et al., 2013). isoWATFLOOD is an extension of the WATFLOOD hydrological model, whereby water and oxygen-18 are simultaneously budgeted throughout the modelled hydrologic cycle. WATFLOOD is a distributed model that uses grouped response units





(GRUs) to simulate streamflow in hydrologically-distinct land cover units (Kouwen et al., 1993; Kouwen, 2014). Process representation within WATFLOOD is considered to be a combination of both conceptual and physical, as certain algorithms are conceptually-based (e.g., evaporation and snowmelt), while others are more based in physics (e.g., channel routing). Due to the coupling of isotopes to each hydrological processes simulated in WATFLOOD, simulation of isotopic composition

does not introduce any additional parameters. A more comprehensive description of isoWATFLOOD's model structure and governing equations can be found in Stadnyk et al. (2013) and select descriptions are provided in Table 6.

isoWATFLOOD requires $\delta^{18}$O of rainfall and snowfall and hourly distributed relative humidity to force the model. Additionally, $\delta^{18}$O compositions for hydrologic storages of river/fen water, soil water, baseflow, and snowpack are needed

for model initialization, which can be obtained from field data or estimated. Here, regional isotopic initializations are derived from measured data (St Amour et al., 2005), and are summarized in Table 5. Sensitivity analyses have shown that within one month of simulation isoWATFLOOD spin-up is complete and, past this point, initialization values have no bearing on model output. All other data required by isoWATFLOOD (e.g., distributed precipitation, temperature, evaporation, inflows, etc.) are passed from WATFLOOD forcings or computations.

The watershed model set-up in this study is based off the version used by Stadnyk et al. (2013), with several changes and improvements. Based on findings from Aylsworth and Kettles (2000), we implemented a 90 % bog and 10 % fen split in Blackstone and a 30 % bog and 70 % fen split in Jean-Marie. The entirety of the FSB is modelled at a 10 km spatial resolution, and the model is run continuously from January 1996 to December 1999; whereby 1996 is utilized as spin-up to

set initial hydrologic and isotopic storage conditions.

### 3.2 Calibration and parameter uncertainty

Being a distributed model, WATFLOOD has a large number of parameters requiring calibration. For this reason, a sensitivity analysis is first conducted to identify which parameters have the largest influence on both streamflow and $\delta^{18}$O$_{SF}$. A subset of parameters are identified for inclusion in the calibration based on this sensitivity analysis, including nine

hydrological parameters from each of the five most prominent land classes (mixed/deciduous, coniferous, transit, bogs and fens), and four routing parameters from each of the two modelled sub-basins. This results in 53 parameters that are incorporated in the parameter uncertainty assessment (Table 6; Table 8). Allowable ranges for each parameter are determined based on published values alongside personal communications with N. Kouwen (Kouwen, 2014) (Table 8).

This study uses a multi-criteria, multi-objective approach to model calibration, with the procedure summarized as follows:

   i.   A Monte Carlo random sampling approach, assuming uniform parameter distributions, is used to individually select each parameter from its allowable range (Table 8). Random parameter sampling is completed 30,000 times, generating 30,000 unique parameter sets for isoWATFLOOD model evaluation.





ii.  For each of the three $\delta^{18}O_{ppt}$ inputs (KPN43, REMOiso and static), streamflow and $\delta^{18}O_{SF}$ are simulated from 1996 to 1999 for all 30,000 parameter sets (as defined in (i)).

iii.  Modelled streamflow and $\delta^{18}O_{SF}$ are assessed statistically over the period of study (1997–1999, excluding the 1996 spin-up year), and regionally across the Jean Marie and Blackstone sub-basins. Simulations are classified as behavioural or non-behavioural based on the following set of efficiency criteria thresholds for streamflow and $\delta^{18}O_{SF}$:

    a.  Streamflow:

        $NSE \geq 0.5$;

        $|\%Dv| \leq 20\%$, and;

        $|\log(\%Dv)| \leq 20\%$.

    b.  $\delta^{18}O_{SF}$:

        $RMSE \leq 2.5$ ‰, and;

        $KGE >= 0.3$.

iv.  All reported results, such as simulation means, percentiles and parameter values, are derived from behavioural simulations corresponding to each $\delta^{18}O_{ppt}$ input. The presented uncertainty bounds are the 5[th] and 95[th] percentiles of simulated streamflow, $\delta^{18}O_{SF}$, and snowpack $\delta^{18}O$, drawn from the behavioural simulations at each time step for each $\delta^{18}O_{ppt}$ input.

The behavioural thresholds used in this study are subjectively defined, however, are arrived at through a review of methods employed in similar studies (e.g., Moriasi et al., 2007; Birkel et al., 2010a; 2010b; 2011; Smith et al., 2016), measurement error, and an iterative process exploring the sensitivity between the set thresholds and resulting behavioural simulations for each input type. Based on this analysis, the Nash-Sutcliffe efficiency (NSE; Nash and Sutcliffe, 1970), volumetric error criteria (%Dv), root mean square error (RMSE), and the Kling-Gupta efficiency criterion (KGE; Gupta et al., 2009; Kling et al., 2012) are selected. A multi-criteria model evaluation approach is used as each criterion places emphasis on different statistical properties of a simulation. For example, NSE has a documented bias towards peak flow, and conversely, log (%Dv) is more appropriate evaluation measure for periods of low flow. The NSE, %Dv, and log(%Dv) efficiency are not considered suitable metrics for $\delta^{18}O_{SF}$ assessment due to the temporal discontinuity of the isotope observations, therefore RMSE and KGE are used as isotopic simulation statistics. It should also be noted that $\delta^{18}O_{SF}$ observations are not equally distributed through time, whereby the highest concentration of observations occurs during snowmelt in the month of May (~25 %), and the fewest observations during the six month ice-on period from November to April (~23 %). The sporadic distribution of observations may result in the calibrations more highly weighted to certain periods of the year and the dominate processes occurring at that time; therefore having the potential to impact model parameterization.



## 4 Results and discussion

Results of the three calibrations indicate that $\delta^{18}O_{ppt}$ input influences the number simulations that meet the behavioural criteria thresholds. The KPN43 input results in an increased number of behavioural simulations (n = 321) in comparison to the REMOiso (n = 268) or static (n = 216) input types (Table 7). This suggests that the choice of $\delta^{18}O_{ppt}$ input may potentially impact internal apportionment of water (i.e., the modelled proportion of water entering, stored, and released from the ground surface, upper and lower zones) through model parameterization. A summary of behavioural parameter set characteristics is provided in Table 8. Land cover parameters are reported as weighted averages for the modelled region.

Among input types there are potentially significant differences in several parameters, which will be explored further throughout the remainder of the manuscript. In almost all instances, the range of the parameters was not significantly constrained from the allowable parameter range. Due to the wide range of behavioural parameter values (Table 8), we are confident that the approach used is sufficient to characterize parameter uncertainty. However, and not unexpectedly, this finding also indicates that 30,000 model evaluations are not sufficient to quantify parameter identifiability for isoWATFLOOD.

### 4.1 Precipitation oxygen-18 input

Time series of $\delta^{18}O_{ppt}$ inputs for Jean Marie and Blackstone are displayed on panel (a) of Fig. 3 and Fig. 4, respectively. On average, KPN43 input is the most enriched (-20.48 ‰), followed by REMOiso (-21.78 ‰), with static being the most depleted (-22.82 ‰). KPN43 and static inputs have similar variation about their mean values, with CV's equal to 0.19 and 0.20, respectively. Conversely, REMOiso has a higher CV (0.25) and much larger range, which is, in part, due to the finer daily time step of this input. Spatial variability between Jean Marie and Blackstone sub-basins is zero for the static input; however some variation among sub-basins is seen for KPN43 and REMOiso. Interestingly, both the KPN43 and REMOiso inputs show, on average, more depleted $\delta^{18}O_{ppt}$ values within Blackstone (-20.79 ‰ and -22.01 ‰, respectively) in comparison to Jean Marie (-20.17 ‰ and -21.54 ‰, respectively), in addition to increased variability. This is likely caused by the higher elevations present in the headwaters of the Blackstone relative to the Jean Marie (a maximum difference of ~215 m).

Although there are only 19 rainfall $\delta^{18}O$ observations collected over the study period for Jean Marie, and eight within Blackstone (hollow black diamonds on Fig. 3 and Fig. 4), these limited data provide some information regarding accuracy of the estimated $\delta^{18}O_{ppt}$ inputs. By visual inspection, each of the three inputs produces reasonable estimates of $\delta^{18}O_{ppt}$. This is expected for the static input as the seasonal compositions are derived from observations; however, this comparison provides some level of validation for KPN43 and REMOiso. REMOiso is the only input that can somewhat replicate the event-scale variability in $\delta^{18}O_{ppt}$ due to its daily time step. Both KPN43 and static inputs appear to generally capture the average





magnitude of summer rainfall events and overall seasonal variability; however there are insufficient observations to statistically support this statement within the Fort Simpson basin.

## 4.2 Modelling streamflow

All calibrations adequately capture variations in total streamflow in both sub-basins, as emphasised by the regional calibration statistics (Table 7). On average, behavioural streamflow simulations have a NSE of 0.68, and %Dv of 13.8 %. Mean daily streamflow and uncertainty bounds for the KPN43, REMOiso and static model calibrations are displayed on panel (b) of Fig. 3 and Fig. 4. Differences in hydrograph characteristics between Jean Marie and Blackstone are due to variations in basin physiography, storage mechanisms and land cover composition, specifically large differences in average basin slope and wetland dynamics (St Amour et al., 2005).

Within the Jean Marie sub-basin, both the timing and volume of peak flows derived from snow melt and early summer rains are well captured in 1998, however, volume is under predicted in 1997 and 1999 for the average streamflow simulation. The model also has difficultly capturing the volume of the recession limb, which may be attributed to the parameterization of baseflow and fen response in this sub-basin. In the Blackstone, the recession limb of the hydrograph and low flow volume are well modelled, however peak flows (with the exception of the 1997 snow melt) are under estimated. Within both sub-basins, flows from 1997 fall rain events are well captured, except for an October 1998 rainfall event that generated a limited streamflow response in both sub-basins. This may point to inadequate precipitation forcing due to the climate station proximity and high spatial variability of rainfall, or possibly inadequate soil moisture parameterization.

An interesting finding is the similarity of mean streamflow simulation despite contrasting $\delta^{18}O_{ppt}$ inputs. Kendall's tau coefficient ($\tau$) is used to determine the level of correlation between streamflow simulations generated by the three $\delta^{18}O_{ppt}$ input methods (and associated model parameterizations). In Jean Marie, $\tau$ ranges between 0.92 (REMOiso versus static) to 0.97 (KPN43 versus static). In Blackstone $\tau$ is more tightly constrained, ranging from 0.96 (REMOiso versus static) to 0.98 (KPN43 versus static). All $\tau$ values are statistically significant. It should be noted that small deviations between mean streamflow simulations occur during spring melt, where REMOiso driven streamflow consistently shows higher peaks than KPN43 and static driven simulations. However, these differences in mean streamflow fall within overlapping uncertainty bounds and therefore are not deemed significant outside of parameter uncertainty.

## 4.3 Modelling $\delta^{18}O$

Mean daily $\delta^{18}O_{SF}$ simulations and uncertainty bounds for KPN43, REMOiso and static model calibrations are displayed on panel (c) of Fig. 3 and Fig. 4. Each calibration produces mean simulations that capture many of the trends (but not particularly the magnitudes) present in the observed $\delta^{18}O_{SF}$ record. Observed $\delta^{18}O_{SF}$ show depletion in streamflow $\delta^{18}O$ due to large influxes of snowmelt during the spring freshets, with $\delta^{18}O_{SF}$ gradually enriching over the summer months due to the





influence of evaporative enrichment of surface and soil waters, occasionally punctuated by rainfall events that may enrich or deplete $\delta^{18}O_{SF}$. During late fall and throughout the winter, $\delta^{18}O_{SF}$ tends toward a more depleted, stable groundwater composition (St Amour et al., 2005).

5   Though each of the calibrations demonstrate many of the same trends as the observed $\delta^{18}O_{SF}$ record, there are notable differences. As simulated $\delta^{18}O_{SF}$ uncertainly envelopes associated with each input type are, at times, non-overlapping, differences in $\delta^{18}O_{SF}$ simulations can be attributed to $\delta^{18}O_{ppt}$ input and therefore are not just an artefact of parameter uncertainty. The dissimilarities between $\delta^{18}O_{SF}$ simulations are also reflected in the RMSE statistic (Table 7). The RMSE is larger for static-derived $\delta^{18}O_{SF}$ simulations due to increased emphasis on periods with observed data (i.e., spring freshet), where larger offsets between simulated and observed $\delta^{18}O_{SF}$ exist. The KGE statistic does not mirror RMSE, as it shows only minor differences between $\delta^{18}O_{SF}$ simulations. The nature of the KGE statistic is to put less emphasis on error offsets derived from peak flows (i.e., spring freshet) by providing a more balanced approach where error is summed first and squared at the end, preserving the sign of the error and enabling a trade-off of error throughout the simulation (Gupta et al., 2009). Therefore, this statistic better reflects the fit of the overall simulation throughout the study period, however, further research is required to better understand the impacts of sporadic sampling resolution (of $\delta^{18}O_{SF}$ observations) on efficiency criteria, and consequently the objective functions(s).

Differences in $\delta^{18}O_{SF}$ simulations within each sub-basin are due to a combination of: (1) the markedly different $\delta^{18}O_{ppt}$ compositions between inputs entering the system during large precipitation events, and (2) the way in which new water flushes through the system via the various hydrological compartments. For this study area, large precipitation events can be further separated into: (1) major rainfall events occurring in post-freshet (summer and fall) months, and (2) the accumulation of winter snowfall and corresponding spring freshet

Post-freshet $\delta^{18}O_{SF}$ simulations are impacted by rainfall amount and composition, as well as the offset between simulated $\delta^{18}O_{SF}$ and $\delta^{18}O_{ppt}$ input at the time of rainfall. As rainfall amount and/or the offset increases, the resulting impact on simulated $\delta^{18}O_{SF}$ also increases. This highlights the importance of capturing the variability in the $\delta^{18}O_{ppt}$ input, particularly for large and isotopically distinct (from streamflow) rainfall events. The threshold defining a large rainfall event will vary depending on factors such as basin physiography, land cover, storage capacity and antecedent conditions. St Amour et al. (2005) estimate this threshold to be ≥40 mm within the Fort Simpson region. An example of a large, yet isotopically distinct, rainfall event is June 11–12, 1998 when approximately 70 mm fell over two days with an observed $\delta^{18}O_{ppt}$ composition of -22.7 ‰. Both REMOiso and static inputs reasonably captured the event (-20.9 ‰ and -20.1 ‰, respectively), however, the KPN43 input predicted a composition of -17.6 ‰. This single event resulted in a significant offset between KPN43 $\delta^{18}O_{SF}$ compared with REMOiso and static $\delta^{18}O_{SF}$ which was maintained throughout the remainder of 1998, until the 1999 freshet.



Throughout much of Canada and other regions experiencing a high-latitude climate, a substantial portion of annual streamflow (and typically the peak flow) is generated during the spring freshet when the accumulation of solid precipitation from the winter season melts over the period of a few weeks. Therefore, it is important to understand how differences in $\delta^{18}O_{ppt}$ input impact snowpack and snowmelt isotopic compositions in isoWATFLOOD. Figure 5 shows the evolution of

precipitation-weighted snowpack oxygen-18 ($\delta^{18}O_{SNW}$) throughout each winter season of the study period.

Comparison of like-forcing pairs between Jean Marie and Blackstone reveals subtle spatial differences in simulated $\delta^{18}O_{SNW}$. There are, however, significant differences between KPN43, REMOiso and static snowpack compositions within each sub-basin. Interestingly, REMOiso and KPN43 simulations of snowpack show similar end of winter precipitation-weighted

$\delta^{18}O_{SNW}$, differing by less than 0.5 ‰ in 1997–1998 and 1998–1999. REMOiso and KPN43 inputs also consistently generate significantly more enriched snowpack compositions throughout the study period in comparison to static $\delta^{18}O_{SNW}$ input. On average, KPN43 is 3.3 ‰ more enriched, and REMOiso is 3.1‰ more enriched than end of season static $\delta^{18}O_{SNW}$. These differences may also stem from insufficiencies in modelled snowpack due to fractionation during sublimation, melting and refreezing of the snowpack that is unaccounted for in the current isoWATFLOOD model. The static input may inadvertently

account for some of these processes, as the specified compositions are from snow pack sampling conducted towards the end of winter (in late March). Research into defining snow fall, pack and melt offsets (from field studies), and refining isoWATFLOOD's cryospheric dynamics and processes is currently ongoing.

These significant differences in simulated snowpack composition are one of the primary causes for the offsets between

KPN43, REMOiso and static $\delta^{18}O_{SF}$ simulations (Fig. 3 and Fig. 4). Once a $\delta^{18}O_{SF}$ simulation has been offset, it is not possible to 'reset' the isotopic composition in late fall when streamflow decreases to near-zero since there is still mass remaining in the system. This can result in compounding isotopic error over a continuous simulation period, thus highlighting the sensitivity of this tracer as a calibration tool. This compounding error is also observed for rainfall events, but generally to a lesser extent due to the relatively smaller durations and magnitudes (volume contributions) of most rainfall

events in high latitude regions.

Provided that both $\delta^{18}O_{SF}$ and $\delta^{18}O_{SNW}$ are significantly different among $\delta^{18}O_{ppt}$ inputs, internal water apportionment (determined by model parameterization) may also be influenced by $\delta^{18}O_{ppt}$ input type. Therefore, hydrograph component contributions are further explored to determine the effect that the differences in $\delta^{18}O_{ppt}$ input has on these contributions.

**4.4 Hydrograph component analysis and parameter distributions**

Percent of volume contributing to total streamflow from surface runoff, interflow and baseflow storage for each season (DJF: December-January-February; MAM: March-April-May; JJA: June-July-August; and, SON: September-October-November) and each of the three $\delta^{18}O_{ppt}$ inputs are shown on Fig. 6. Jean Marie and Blackstone sub-basins generally display similar





trends in internal water apportionment throughout the year, indicating generally similar model parameterizations. Some seasonal differences are visible, which can be linked to variations in basin physiography, land cover, and storage characteristics reflected by differences in the baseflow (lzf and pwr) and wetland parameters (kcond and theta) between basins (Table 8). Overall, the freshet and post-freshet percent volume contributions to total streamflow in this study are in

general agreement with those reported in previous studies. For example, St Amour et al. (2005) also found groundwater contributions to be significant throughout the year, and estimated post-freshet contributions to total streamflow at 71 % (± 9 %) and 64% (± 10 %) within Jean Marie and Blackstone, respectively. Snowmelt contributions were estimated to be 21 % (± 2 %) and 40 % (± 4 %) of total streamflow volume for Jean Marie and Blackstone. Additionally, Jasechko et al. (2016) estimate that annually 80 – 90 % of the Mackenzie River streamflow is "old" water (i.e., water that has not entered the

stream within the last 2.3 ± 0.8 months). Their findings also suggest that the annual percentage of old streamflow can be higher in mountainous watersheds with steeper slopes, such as the FSB, than lower-gradient watersheds.

Comparison of seasonal volume contributions derived from each $\delta^{18}O_{ppt}$ input reveal that during spring (MAM), REMOiso-driven simulations show significantly more surface flow contribution to total streamflow, with the mean volume lying above

15 the 95[th] percentile for both the KPN43 and static input simulations (Fig. 6). On average, REMOiso simulations contribute almost twice as much surface runoff to total streamflow as KPN43 and static simulations during MAM (39 % versus 25 % and 22 %, respectively, for Jean Marie; and similar, yet slightly larger, percent contributions for Blackstone).

Based on the averaged seasonal analysis, no other significant differences in component contributions outside of parameter

uncertainty can be attributed to $\delta^{18}O_{ppt}$ input. It is important to note, however, that each $\delta^{18}O_{ppt}$ input results in differing amounts of parameter uncertainty, both seasonally and overall, as represented by differing widths of uncertainty bounds (cross symbols) on Fig. 6. The variation in uncertainty bounds between $\delta^{18}O_{ppt}$ inputs is also visible on Fig. 3 through Fig. 5. The REMOiso input yields the largest amount of uncertainty in total streamflow, also reflected in the relatively larger amounts of uncertainty in surface water and baseflow component contributions (Fig. 6). Conversely, KPN43 and static

inputs generate similar or slightly larger uncertainty in interflow (soil water) contributions relative to REMOiso and lower uncertainty surrounding surface and baseflow contributions, and overall total streamflow. These differences in uncertainty are attributed to the number, and characteristics of the behavioural parameters retained for each $\delta^{18}O_{ppt}$ input, which originate due to distinctions in magnitude and variability (both spatial and temporal) among $\delta^{18}O_{ppt}$ inputs.

Further demonstrated by parameter probability distributions (Fig. 7), the three calibrations resulted in noteworthy differences in behavioural parameters. We do not display these distributions to comment definitively on parameter identifiability because, as previously noted, the number of evaluations was insufficient for that purpose. But rather, we introduce this analysis for select parameters to reinforce and explain the findings from Fig. 3 through Fig. 5, and to highlight that within this study, model parameterization is impacted by $\delta^{18}O_{ppt}$ input. The selected parameters influence evaporation (f-ratio),





surface runoff during snowmelt (akfs, base), upper and lower zone storage (retn), interflow (retn), and baseflow (lzf, pwr). Results show that more often than not, REMOiso parameter distributions are different than KPN43 or static parameter distributions. Although there are dissimilarities between KPN43 and static parameter distributions, however these are typically not as prevalent. This echoes the findings from Fig. 7 that KPN and static derived contributions to total streamflow are more similar than contributions arising from REMOiso; which may very well be an artefact of the increased temporal and spatial variability in the REMOiso $\delta^{18}O_{ppt}$ input relative to that of the KPN43 and static inputs.

Differences in surface water contribution during snowmelt between REMOiso, KPN43 and static inputs are likely be explained by differences in the akfs and base parameters. Parameter distributions derived from REMOiso are significantly different (as verified through Kolmogorov–Smirnov testing of distributions) than those from the KPN43 and static inputs for these parameters (Fig. 7, panels (b) and (f)). Lower akfs values represent decreased infiltration and increased surface runoff during snowmelt, which corresponds to REMOiso's increased surface water contributions to total streamflow during spring (MAM). Differences in baseflow contribution and uncertainty between $\delta^{18}O_{ppt}$ inputs are attributed, in part, to differences in the lzf and pwr parameters (Fig. 7, panels (c-d) and (g-h)), which have a large impact on the quantity of baseflow and the slope of the recession limb of the hydrograph. Wider uncertainty bounds for REMOiso relative to KPN43 and static within Blackstone (Fig. 6, panel (f)), and for all $\delta^{18}O_{ppt}$ inputs during fall and winter within Jean Marie (Fig. 6, panel (c)), are likely due to the wider range of behavioural values for the pwr parameter, specifically the inclusion of lower values which results in longer, more drawn out recession limbs.

Although more work is required towards assessing and understanding parameter identifiability for WATFLOOD, the above analysis shows that selection of $\delta^{18}O_{ppt}$ input has direct implications on model parameterization, and this source of uncertainty should be considered in future studies.

## 5 Conclusions and recommendations

This study uses three types of estimated $\delta^{18}O_{ppt}$ input to force a tracer-aided hydrological model, isoWATFLOOD, and investigates the impact that $\delta^{18}O_{ppt}$ inputs of differing spatial and temporal variability have on total streamflow, isotopic composition of streamflow, and the seasonality of individual hydrograph components. This work informs the over-arching goal of quantifying and reducing uncertainty (and equifinality) in isoWATFLOOD streamflow simulations.

This study demonstrated that although total simulated streamflow is not significantly affected by choice of $\delta^{18}O_{ppt}$ input, $\delta^{18}O_{SF}$ simulations and the internal apportionment of water (surface flow, interflow, and baseflow contributions) in WATFLOOD can be significantly impacted, especially during large precipitation and snowmelt events. The ability of estimated $\delta^{18}O_{ppt}$ to capture both the variability and seasonality in precipitation isotopes, especially when precipitation events





display distinctly different isotopic compositions than that of streamflow, is critical for tracer-aided model forcing. Differences in $\delta^{18}O_{SF}$ and water partitioning between compartments are driven by differences in model parameterization, as witnessed by variations in the amount of uncertainty and parameter distributions between $\delta^{18}O_{ppt}$ input types. This suggests that choice of $\delta^{18}O_{ppt}$ input impacts parameterization of WATFLOOD, and for this reason, if estimates of $\delta^{18}O_{ppt}$ are used in

modelling, modellers should account for this input uncertainty in overall uncertainty assessments. Findings also show that simulations of total streamflow did not show significant differences between the three $\delta^{18}O_{ppt}$ inputs and corresponding parameterizations, despite $\delta^{18}O_{SF}$ simulations displaying significant differences. This reinforces to the utility of tracer-aided models to diagnose issues with equifinality in model simulations.

As WATFLOOD is a complex model with a large amount of parameters, it is important to work towards conducting a comprehensive study focusing on $\delta^{18}O_{ppt}$ input uncertainty and parameter identifiability. Ideally, further studies should be conducted in a watershed that is adequately instrumented to characterize observed $\delta^{18}O_{ppt}$ input both spatially, but more importantly, temporally (i.e., daily or weekly sampling resolution). This will facilitate a 'baseline' model calibration (using observed $\delta^{18}O_{ppt}$) from which deviations in parameter distributions due to estimated $\delta^{18}O_{ppt}$ input can be more intensely

explored. This type of study would allow further investigation of several key questions: first, if these pseudo-forcings are adequate alternatives in place of $\delta^{18}O_{ppt}$ observations; second, if there is a specific subset of model parameters that are more sensitive to estimated $\delta^{18}O_{ppt}$ input, and how (if at all) these parameters compensate for compounding error stemming from estimated $\delta^{18}O_{ppt}$ input. Unfortunately, at least within Canada, a well instrumented watershed at the regional scale does not yet exist. This again points to the importance of implementing additional (or enhancing current) hydro-meteorological

monitoring networks. If observation networks allow, this type of study should also be conducted in watersheds of differing dominant hydrological processes (e.g., rainfall-dominated versus snowmelt-dominated) to better understand $\delta^{18}O_{ppt}$ input uncertainty on parameterization across the range of modelled hydrologic processes.

As expected, REMOiso exhibits some bias and will continue to need correction before application within Canadian

watersheds. More studies are needed to examine the differences between $\delta^{18}O_{ppt}$ observations and REMOiso simulations throughout Canada to better understand the nature of this bias, and the most appropriate bias correction methods; which can be done using observations from the CNIP database. This feedback regarding REMOiso performance across Canada is also beneficial to model developers. Additionally, the suitability and performance of other isotope-enabled RCM's for use in Canada should be explored. Regarding the usefulness of the inputs within regions of limited $\delta^{18}O_{ppt}$ observations, both the

static and REMOiso inputs require existing $\delta^{18}O_{ppt}$ observations (i.e., from CNIP) to either define or bias correct the input, which may limit their use for certain applications. If these data are not available, the KPN43 input provides reasonable results without the need for any additional observations. For all inputs, the existence of CNIP (and other isotopes in precipitation networks) observations is crucial to the development, validation, and bias correction of estimated $\delta^{18}O_{ppt}$ inputs.





Understanding how estimated $\delta^{18}O_{ppt}$ input affects model parameterization and subsequently internal distribution of water is important to understanding overall uncertainty of the model and the model's abilities and limitations without access to observed $\delta^{18}O_{ppt}$ times series data. Canada, alongside many other countries, continues to have significant regions with sparse

hydro-meteorological observations (Coulibaly et al., 2013). This study achieves an understanding of how isoWATFLOOD can be used in regions with a limited number of $\delta^{18}O_{ppt}$ observations, and that the model can be of value in such regions. This study reinforces that a tracer-aided modelling approach assists with resolving hydrograph component contributions, and works towards diagnosing the issue of model equifinality and knowledge that modellers are achieving the 'right answers for the right reasons' (Kirchner, 2006). Attaining this understanding of $\delta^{18}O_{ppt}$ input uncertainty on simulated model output is

especially important when calibrated models are used as tools to assess how changes in climate or land-use effect future predictions of streamflow.

**Author contribution**

C. Delavau developed model code to generate Kpn $\delta^{18}O_{ppt}$ input, perform Monte Carlo simulations, and process the corresponding output. T. Stadnyk and T. Holmes developed and enhanced isoWATFLOOD code for the version of

isoWATFLOOD used in this study. C. Delavau performed the analysis presented in this manuscript, with assistance from T. Stadnyk. C. Delavau prepared the manuscript with contributions from all co-authors.

**Competing interests**

The authors declare that they have no conflict of interest.

**Acknowledgements**

The authors would like to acknowledge Dr. K. Sturm for provision of the REMOiso data utilized in this study. Additional thanks go to Dr. N. Kouwen for direction and input on WATFLOOD modelling. CNIP is made possible through the help of the Canadian Air and Precipitation Monitoring Network (CAPMoN) for sample collection - Kaz Higuchi and Dave MacTavish in particular, the Environmental Isotope Laboratory at the University of Waterloo for sample analysis, and Tom Edwards for initiating and maintaining the network. We would also like to acknowledge Dan McKinney for provision of the

ANUSPLIN data used in this study. Finally, we would like to acknowledge the contributions of our reviewers whose valuable input has improved this manuscript significantly. This research was partially funded by Natural Sciences and Engineering Research Council (NSERC) Alexander Graham Bell Canada Graduate Scholarship (CGS-D).



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





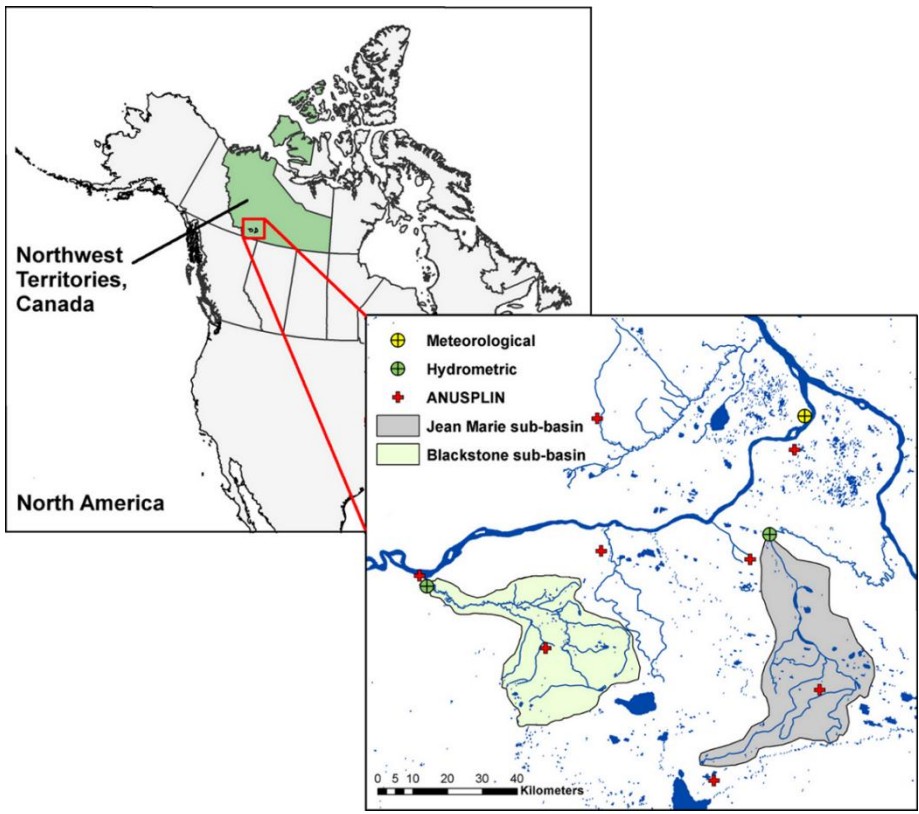

**Figure 1: Fort Simpson River Basin (all other tributaries of the Liard and Mackenzie Rivers have been removed for ease of viewing).**

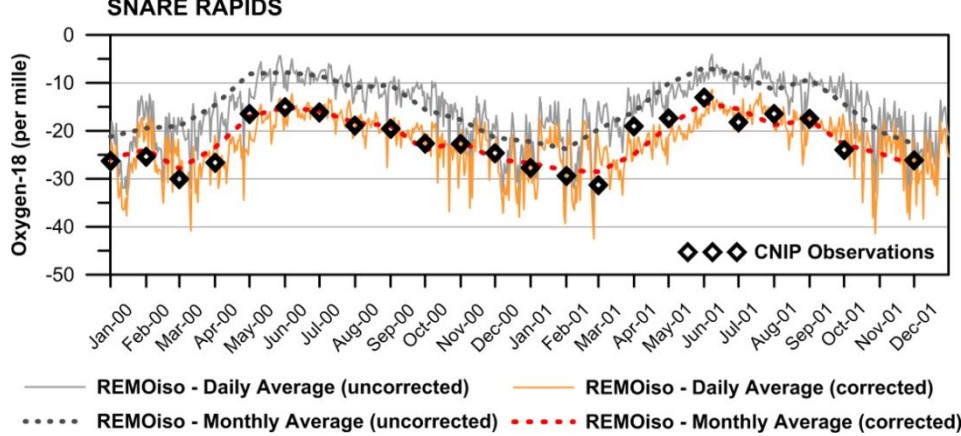

**Figure 2: Comparison of raw and corrected REMOiso $\delta^{18}O_{ppt}$ output with CNIP monthly compositions at Snare Rapids, NWT.**





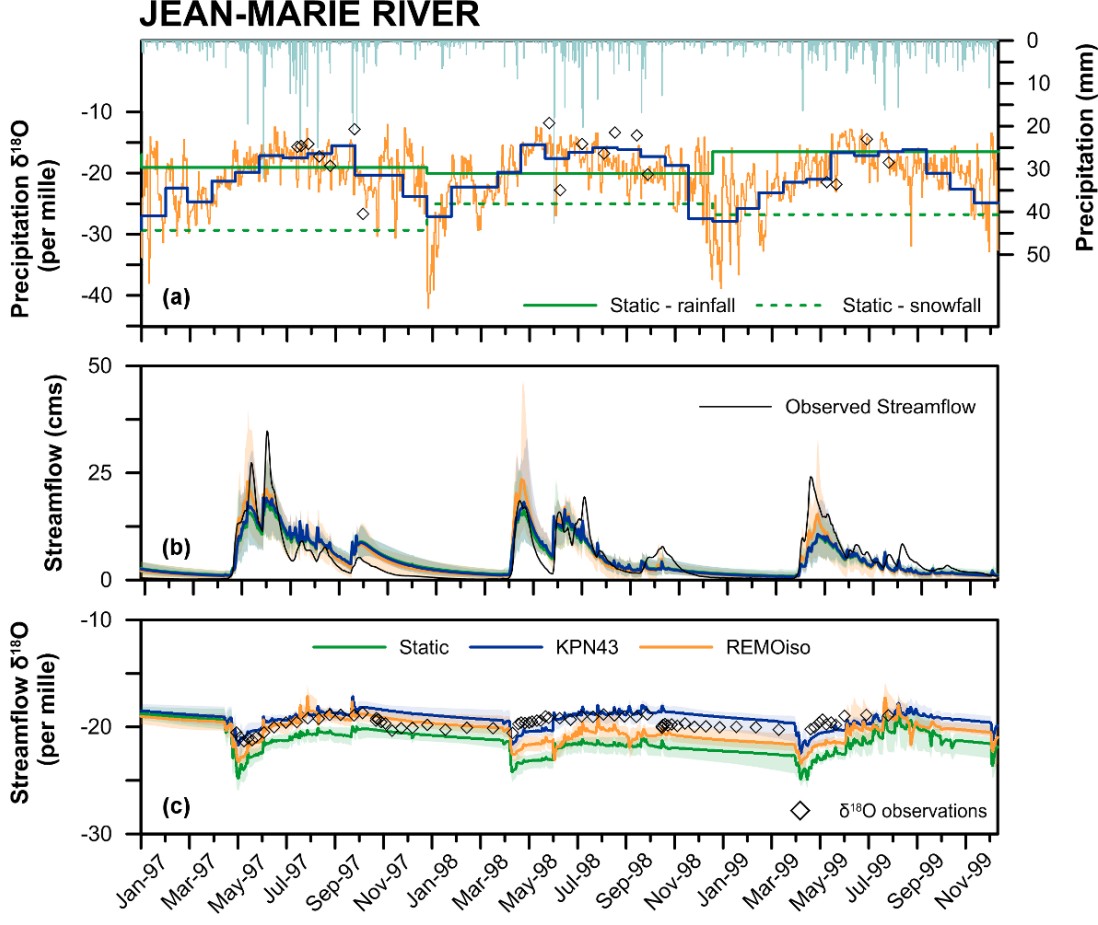

**Figure 3: Input and behavioural simulations for Jean Marie, including: (a) KPN43, REMOiso and static $\delta^{18}O_{ppt}$ input time series and daily precipitation; and simulated (b) mean daily streamflow and uncertainty bounds and (c) mean daily $\delta^{18}O_{SF}$ and uncertainty bounds, for KPN43, REMOiso and static driven model calibrations. $\delta^{18}O_{ppt}$ input-specific uncertainty bounds are represented as the shaded regions, with shading colour corresponding to $\delta^{18}O_{ppt}$ type.**





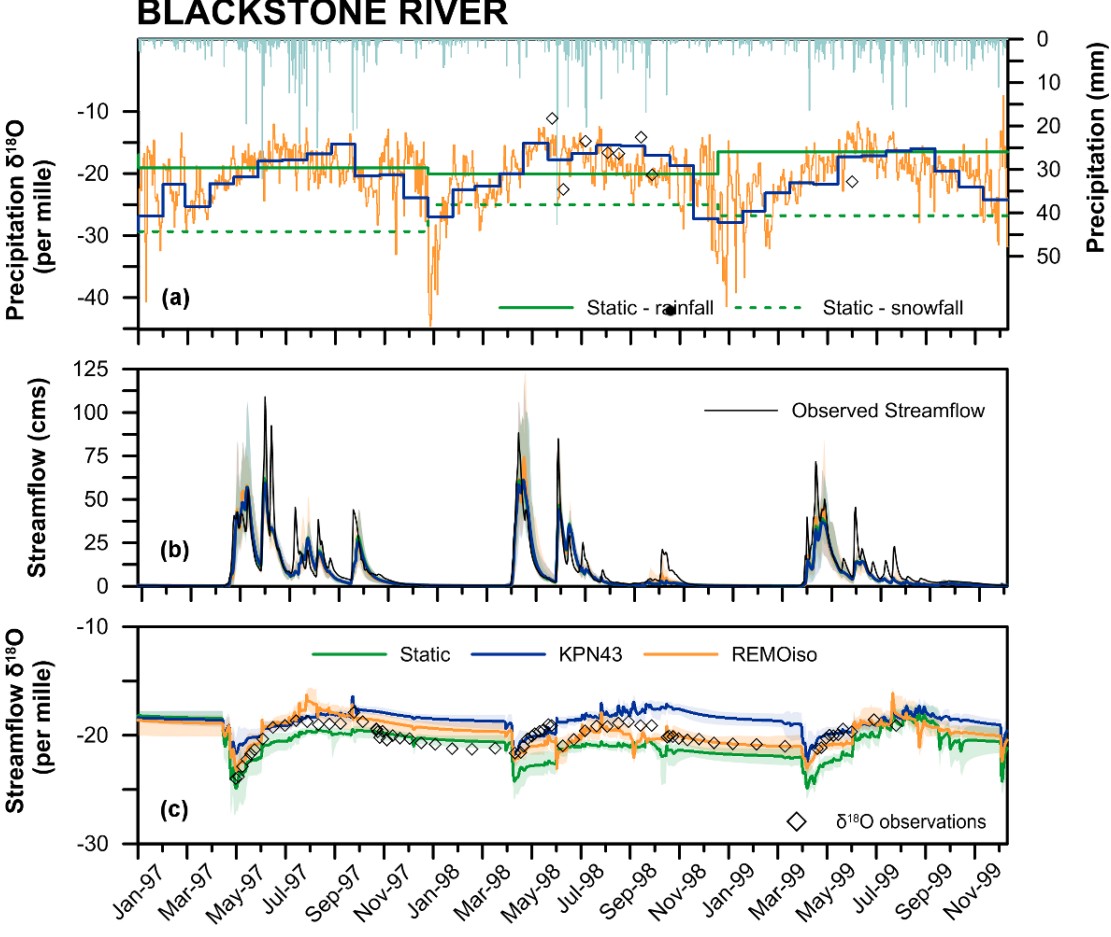

**Figure 4: Input and behavioural simulations for Blackstone, including: (a) KPN43, REMOiso and static $\delta^{18}O_{ppt}$ input time series and daily precipitation; and simulated (b) mean daily streamflow and uncertainty bounds and (c) mean daily $\delta^{18}O_{SF}$ and uncertainty bounds, for KPN43, REMOiso and static driven model calibrations. $\delta^{18}O_{ppt}$ input-specific uncertainty bounds are represented as the shaded regions, with shading colour corresponding to $\delta^{18}O_{ppt}$ type.**





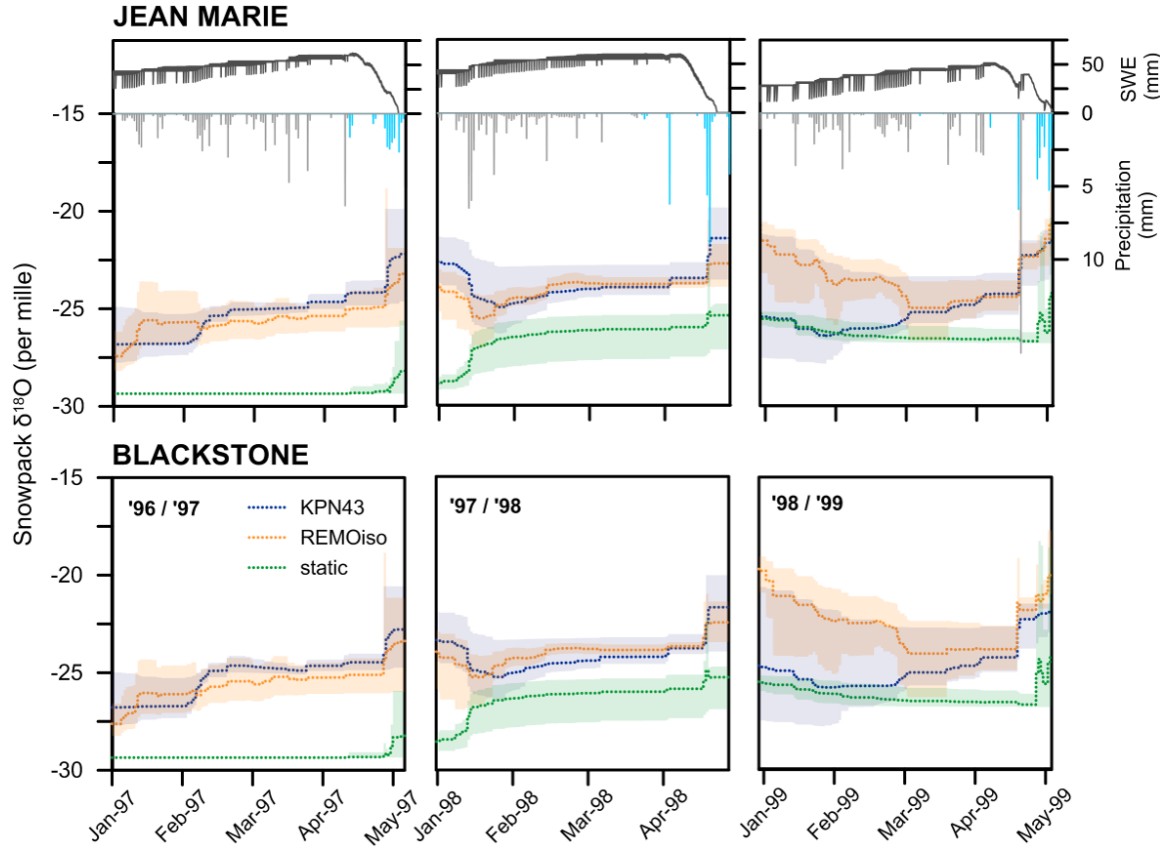

**Figure 5: Precipitation-weighted δ¹⁸O of snowpack (δ¹⁸O$_{SNW}$) for KPN43, REMOiso and static inputs from January to the end of melt for each year of the study period. Snow water equivalent (SWE), snowfall (gray line) and rainfall (blue line) are also shown. δ¹⁸O$_{ppt}$ input-specific uncertainty bounds are represented as the shaded regions.**





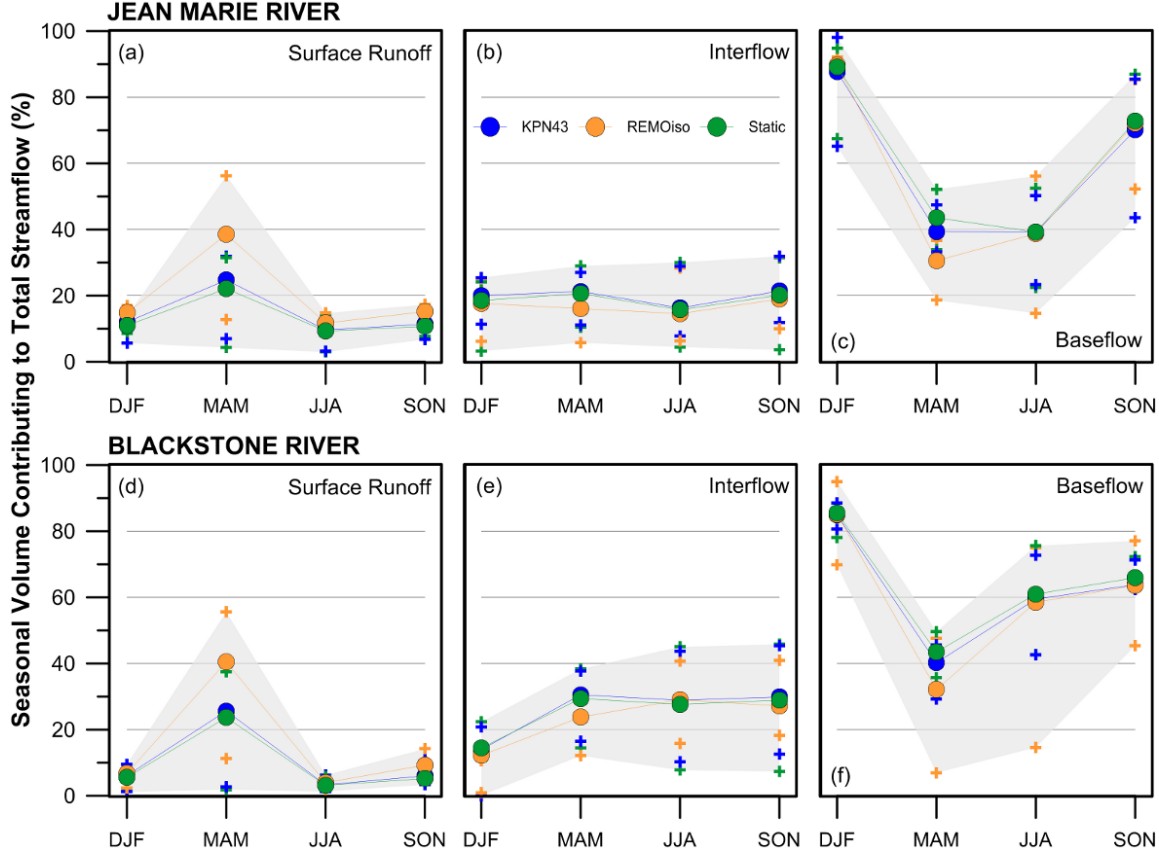

**Figure 6: Percent seasonal volume contributing to total streamflow from surface runoff, interflow and baseflow storages for each season. Cross symbols represent the 5th and 95th percentiles for each forcing method, and circle symbols signify the mean values. The combined uncertainty bounds representing the 5th and 95th simulations from all three $\delta^{18}O_{ppt}$ input types are shaded in gray.**





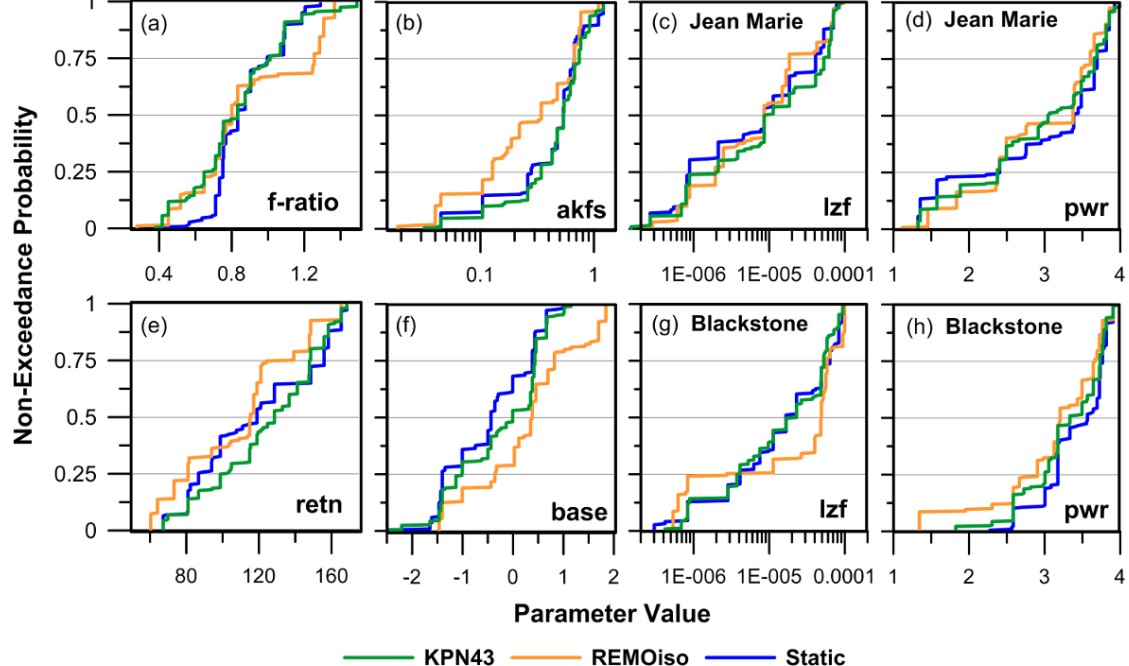

**Figure 7: Probability distributions for select parameters, as indicated in the bottom right corner of each panel. Parameters are from behavioural simulations, and (a), (b), (e) and (f) have been weighted to the land cover distribution within Jean Marie and Blackstone, as outlined in Table 1. Panels (c) and (d) and river class parameters within Jean Marie, and panels (g) and (h) contain river class parameters for Blackstone.**



**Table 1: Basin characteristics, including land cover classification, area, and average basin slope (recreated from data provided in St Amour et al., 2005)**

| Sub-basin | Land Cover Classification (%) | | | | | | Area (km$^2$) | Basin Slope (%) |
|---|---|---|---|---|---|---|---|---|
| | Deciduous | Mixed | Coniferous | Transitional | Wetland | Water | | |
| Jean- Marie River | 5 | 22 | 23 | 31 | 14 | 1.3 | 1310 | 0.3 |
| Blackstone River | 7 | 17 | 14 | 39 | 21 | 0.7 | 1390 | 0.63 |





**Table 2: Static $\delta^{18}O_{ppt}$ input compositions of annual rainfall and snowfall oxygen-18 for isoWATFLOOD.**

| Year | $\delta^{18}O$ rainfall (‰) | $\delta^{18}O$ snowfall (‰) |
|------|------|------|
| **1996** | -17.00 | -29.35 |
| **1997** | -19.10 | -29.35 |
| **1998** | -20.10 | -25.03 |
| **1999** | -16.52 | -26.79 |





**Table 3: Seasonal REMOiso bias correction.**

| Season | Correction (‰) |
|---|---|
| **November – December – January – February** | -4.5 |
| **March – April – May** | -8.9 |
| **June – July – August** | -7.3 |
| **Septemer – October** | -8.5 |
| **Average Correction Applied:** | -7.0 |





**Table 4: Data summary for the study period (SP) and period of record (PoR). The coefficient of variation (CV) is calculated as the ratio of the standard deviation to the mean.**

| Variable (gauge ID) | Unit | Number of Measurements | Mean (SP, PoR) | CV (SP, PoR) | SP Range (min, max) |
|---|---|---|---|---|---|
| **Hydrometric/Meteorological Data** | | | | | |
| **Daily Average Streamflow Jean Marie (10FB005)** | $m^3/s$ | 1095 | 4.66, 5.25 | 1.24, 2.06 | 0.19, 34.9 |
| **Daily Average Streamflow Blackstone (10ED007)** | $m^3/s$ | 1095 | 8.96, 10.76 | 1.65, 2.17 | 0.04, 109 |
| **Mean Daily Air Temperature Fort Simpson (2202101)** | °C | 1093 | -1.5, -3.02 | N/A | -40.8, 25.3 |
| **Daily Precipitation Fort Simpson (2202101)** | mm | 1088 | 1.12, 1.01 | 3.04, 3.19 | 0.0, 43.0 |
| **Hourly Relative Humidity\* Fort Simpson (2202101)** | % | 26280 | 73.9 | 0.24 | 14, 100 |
| **Isotopic Measurements\*** | | | | | |
| **Streamflow $\delta^{18}O$ - Jean Marie** | ‰ | 71 | -19.70 | 0.03 | -21.34, -18.72 |
| **Streamflow $\delta^{18}O$ - Blackstone** | ‰ | 69 | -20.17 | 0.06 | -24.01, -17.92 |
| **Rainfall $\delta^{18}O$ Jean Marie and Blackstone** | ‰ | 27 | -17.55 | 0.23 | -26.70, -11.12 |
| **Precipitation $\delta^{18}O$ Forcing\*** | | | | | |
| **KPN43 $\delta^{18}O_{ppt}$ input** | ‰ | 1800 (36 values at 50 grid points) | -20.48 | 0.19 | -28.86, -13.91 |
| **REMOiso $\delta^{18}O_{ppt}$ input** | ‰ | 54750 (1095 values at 50 grid points) | -21.78 | 0.25 | -42.82, -10.68 |
| **Static $\delta^{18}O_{ppt}$ input** | ‰ | 300 (6 values at 50 grid points) | -22.82 | 0.20 | -29.35, -16.52 |

\* Provided only for the study period, 1997 – 1999.





**Table 5: Initialization values for FSB isoWATFLOOD simulations.**

| Variable | Description | Value (‰) |
|---|---|---|
| $\delta^{18}O_{sf}$ | Background delta for river water initialization | -13.52 |
| $\delta^{18}O_{IF}$ | Background delta for soil water initialization | -14.60 |
| $\delta^{18}O_{GW}$ | Background delta for groundwater initialization | -20.00 |
| $\delta^{18}O_{SNW}$ | Background delta for snow initialization | -22.00 |





**Table 6: Parameters included in the Monte Carlo calibration, alongside a description of what the parameter represents and the algorithm it is used within.**

| Name | Description | Algorithm |
|---|---|---|
| **Routing Parameters** | | |
| **flz** | Lower zone drainage function | An exponential ground water depletion function that gradually diminishes the base flow. Ground water is replenished by drainage of the UZS: |
| **pwr** | Lower zone drainage function exponent | $QLZ = LZF*(LZS)^{PWR}$<br>Where: LZS is lower zone storage<br>QLZ is the baseflow flux |
| **theta** | Wetland porosity | Physically-based wetland routing algorithm |
| **kcond** | Conductivity parameter | (McKillop et al., 1999) |
| **Hydrologic Parameters** | | |
| **f-ratio** | Interception capacity multiplier | Conceptual evaporation algorithm based on Hargreaves and Samani (1982). f-ratio is a multiplier for the interception capacity for each land class. |
| **ak** | Surface permeability (bare ground) | Conceptual infiltration algorithm (similar to Green and Ampt, 1911); but based on Richard's equation which is physically-based (Philip, 1954) |
| **akfs** | Surface permeability | |
| **rec** | Interflow coefficient | Interflow is represented by a simple storage-discharge relation:<br>$DUZ = REC * (UZS-RETN)*Si$ |
| **retn** | Upper zone retention [mm] | Where: UZS = upper zone storage<br>DUZ = depth of upper zone storage released as interflow<br>Si = internal land surface slope |
| **ak2** | Recharge coefficient (bare ground) | Upper zone to lower zone drainage is represented by a simple storage-discharge relation:<br>$DRNG = AK2 * (UZS - RETN)$<br>Where: DRNG is the drainage from UZS to LZS |
| **mf** | Melt factor [mm/°C/hr] | $M = MF (T_a - base)$ |
| **base** | Base Temperature [°C] | Anderson (1976) |
| **sub** | Sublimation factor | Sublimation is modelled by a static sublimation factor. Amount of sublimation is a fraction of the observed snowfall. For new model setups, the sublimation factor has been replaced by a static sublimation rate. |





**Table 7: Average simulation statistics from n behavioural simulations for streamflow and $\delta^{18}O_{SF}$ for the three model calibrations (using KPN43, REMOiso, and static inputs).**

| Average statistics from n behavioural simulations | KPN43 | REMOiso | Static |
|---|---|---|---|
| **n** | 321 / 30000 | 268 / 30000 | 216 / 30000 |
| **Streamflow (1095 observations for performance evaluation)** | | | |
| **NSE** | 0.68 | 0.68 | 0.69 |
| **\|% Dv\|** | 13.9 | 13.4 | 14.2 |
| **\|Log(% Dv)\|** | 11.5 | 8.9 | 11.6 |
| **$\delta^{18}O_{SF}$ (140 observations for performance evaluation)** | | | |
| **RMSE (‰)** | 1.39 | 1.32 | 2.09 |
| **KGE** | 0.36 | 0.33 | 0.35 |





**Table 8: Allowable parameter ranges and resulting averaged parameter characteristics for behavioural simulations retained from the three model calibrations (KPN43, REMOiso, and static inputs). Parameters are summarized as: median (minimum, maximum).**

| Parameter | Allowable Range | KPN43 | REMOiso | static |
|---|---|---|---|---|
| | | **Jean Marie Sub-basin** | | |
| **flz** | $[1\times10^{-7}, 1\times10^{-4}]$ | $8.60\times10^{-6}$ $(1.49\times10^{-7}, 9.32\times10^{-5})$ | $8.60\times10^{-6}$ $(2.03\times10^{-7}, 9.86\times10^{-5})$ | $8.60\times10^{-6}$ $(2.03\times10^{-7}, 9.86\times10^{-5})$ |
| **pwr** | $[1, 4]$ | 3.05 (1.33, 4.00) | 3.38 (1.12, 4.00) | 3.39 (1.12, 3.96) |
| **theta** | $[0.1, 1.0]$ | 0.60 (0.13, 0.96) | 0.61 (0.13, 1.0) | 0.60 (0.13, 0.98) |
| **kcond** | $[0.1, 1.5]$ | 0.79 (0.17, 1.46) | 0.68 (0.17, 1.46) | 0.86 (0.17, 1.46) |
| | | **Blackstone Sub-basin** | | |
| **flz** | $[1\times10^{-7}, 1\times10^{-4}]$ | $2.30\times10^{-5}$ $(4.19\times10^{-7}, 9.41\times10^{-5})$ | $4.91\times10^{-5}$ $(5.36\times10^{-7}, 9.95\times10^{-5})$ | $1.67\times10^{-5}$ $(3.02\times10^{-7}, 9.86\times10^{-5})$ |
| **pwr** | $[1, 4]$ | 3.34 (1.82, 3.96) | 3.21 (1.35, 3.96) | 3.57 (2.28, 3.91) |
| **theta** | $[0.1, 1.0]$ | 0.55 (0.11, 1.0) | 0.60 (0.13, 1.0) | 0.52 (0.24, 1.0) |
| **kcond** | $[0.1, 1.5]$ | 0.69 (0.11, 1.41) | 0.80 (0.21, 1.48) | 0.69 (0.17, 1.49) |
| | | **Land Cover Weighted-Average Parameter results: median (minimum, maximum)** | | |
| **fratio** | $[0.1, 2.5]$ | 0.70 (0.12, 2.22) | 0.70 (0.15, 2.16) | 0.80 (0.12, 2.23) |
| **ak** | $[1, 50]$ | 21.6 (2.1, 47.6) | 25.8 (2.4, 47.5) | 23.6 (1.2, 46.9) |
| **akfs** | $[0.005, 2]$ | 0.212 (.006, 1.878) | 0.059 (0.006, 1.724) | 0.203 (0.006, 1.850) |
| **rec** | $[0.05, 1]$ | 0.47 (0.08, 0.90) | 0.46 (0.08, 0.88) | 0.43 (0.09, 0.90) |
| **retn** | $[10, 200]$ | 122 (18, 189) | 119 (23, 181) | 114 (20, 186) |
| **ak2** | $[.001, 0.2]$ | 0.013 (0.001, 0.188) | 0.008 (0.001, 0.172) | 0.021 (0.001, 0.184) |
| **fm** | $[0.075, 0.2]$ | 0.117 (0.076, 0.189) | 0.119 (0.078, 0.190) | 0.112 (0.076, 0.189) |
| **base** | $[-3.5, 3.5]$ | -0.20 (-3.28, 3.14) | 0.34 (-2.81, 3.13) | -0.35 (-3.11, 2.85) |
| **sub** | $[0.1, 1.1]$ | 0.53 (0.11, 1.05) | 0.43 (0.13, 1.05) | 0.50 (0.11, 1.05) |