# Peer review of "Examining the impacts of precipitation isotope input ( $\delta^{18}O_{ppt}$ ) on distributed, tracer-aided hydrological modelling"

_Hydrology and Earth System Sciences, 2016_

## Referee Comment (RC1) · Anonymous Referee #1 · 26 Nov 2016

Review of hess-2016-539: Examining the impacts of estimated precipitation isotope (delta oxygen-18) inputs on distributed tracer-aided hydrological modelling by Delavau et al.

General comments: The authors present an interesting case study about using differing precipitation stable isotope input datasets for distributed hydrological modeling in the Northwest Territories of Canada. Based on three different precipitation stable isotope datasets, three different calibrations of the model isoWATFLOOD were identified based on a Monte Carlo random sampling approach. The results show that modeled stream-flow was relatively similar for each of the three used stable isotope input datasets. Whereas the differences in the modeled stable isotope signature in streamflow and the

internal apportionment were much more pronounced. However, the study is lacking some important and critical explanation of the presented model outputs. Please find a detailed description of specific comments and technical notes below. The focus of the presented study is in the scope of HESS. Furthermore I will highlight at this point that the paper is very well written, understandable and the sections are mostly very well structured. However, based on my review below I recommend major revisions prior to a publication in HESS.

Specific comments: The authors should rethink the use of the word "estimated" in the title as well as throughout the whole manuscript. It suggests that the input data was generated specifically for the presented study. It should be clear that (2 of 3) available precipitation isotope product were used to the study. Which is actually an asset for the study and with respect to future studies in other basins.

The first sentence of the abstract is "...increasingly popular tools as they have documented utility in constraining model parameter space during calibration, reducing model uncertainty, and assisting with selection of appropriate model structures.". However, there is no evidence for that statement. Please include additional information to the introduction section or revise the first sentence of the abstract.

The authors highlight the importance of snowmelt in the study region. The stable isotope signature of the snow pack and its melt water is a very challenging topic. Please handle this point very carefully in your publication. On page 5, Line 17 for example you mention that the default method for oxygen-18 input is annual average rainfall and snowfall. In your static approach, however, you used average measurements of rainfall and snowpack from the GEWEX campaign. Please provide the values of snow pack stable isotope signature in figure 5 by the way. Especially during the ablation season the isotopic evolution of the snowpack progresses due to percolating rain water and fractionation caused by processes like melting and sublimation (Zhou et al., 2008; Unnikrishna et al., 2002; Dietermann and Weiler, 2013; Lee et al., 2010). This leads to an increase of heavy isotopes in melt water throughout the freshet period (Taylor et

al., 2001,2002; Unnikrishna et al., 2002). Which is correctly represented by the shown model results. Taylor et al. (2001 and 2002) point out that for hydrological applications (in their case isotope based hydrograph separation) a correct representation oft he snow pack melt water is absolutely crucial.

REMOiso is a distributed dataset and the precipitation amounts are also available spatially distributed over the study area. Why was the precipitation amount weighting only conducted at one location and not spatially distributed?

The authors mention that "several changes and improvements" (Page 7, Line 16) were carried out in the model version used for the study. In the following only one modification (proportion of bog an fen split) is mentioned. Are there any other modifications? If so, please mention them here.

The first two paragraphs of section 4 (Results and discussion) should definitively be revised. There is a lot of content that can be mentioned later in the conclusions section (the last sentence on Line 12-14 for example).

In section 4.2 (Modelling streamflow) please explain the model results as well as the observed streamflow in much more detail. The three different inputs (and three different calibrations) provide very similar results for the simulated streamflow (Page 15, Lines 5-8). Those results should be discussed in more detail. Is there really no discharge in winter (Figure 2 and 3)? What are the influences of groundwater on the hydrology of the region? The same holds for section 4.3. Explain the results in more details. There is especially the time of the spring freshet that needs much more carefully discussed. The model results show a sharp drop of streamflow stable isotope signature, while the observed values are getting more and more enriched at that time. This completely opposed development may be related that the contribution of snowmelt water to total streamflow during that time is too high (or the signature of the snow melt signal is wrong, please remind here my suggestions above) and the contribution of baseflow too low. The contributions of baseflow (groundwater) to total streamflow during the

post-freshet are especially for Jean Marie River much lower in the present model study compared to the results of St Amour et al. (2005). Furthermore, please provide the stable isotope signature of groundwater. And compare those observed values with the values generated by the models in the groundwater routine after the spin-up period.

Please check the manuscript for repetitive information. The sentence on Page 13, 34+35 for example appears almost identical on the next page again (Page 14, Lines 20-22). This would be an excellent take-home sentence for the conclusions section by the way.

In general, I am missing some distinct conclusions in the conclusions section of the submitted manuscript. There are a lot of recommendations and speculations but no clear take-home messages.

Technical notes:

Page 1, Line 18: "…to capture both the variability and seasonality". There it would be better to write "spatial variability and seasonality" or "spatial and temporal variability", since the seasonality is also a variability (temporal).

Page 1, Line 31: (e.g. Beven and Binley, 1992; Kirchner….)

Page 3, Line 22 and Line 29: Please provide size and elevation characteristics of the basins here.

Page 3, Line 27: "…is selected based ON data availability."

Page 4, Line 20: The study region is not a high elevation region. Please mention correctly why the approach is suitable for the study region.

Page 4, Line 21: "…is used TO spatially…"

Page 5, Line 9: Why are they not adequate for model forcing? This is the input data used and referred STATIC in the study, right? Please revise this sentence

[Figure]

Page 5, Line 12: "such that" appears twice.

Page 5, Line 24: KP43 instead of KPN43.

Page 6, Line 4: From my point of view the section 2.4.1 is a description of methods and should therefore be moved in the appropriate section.

Page 6, Line 16: Please mention that Snare Rapids is a CNIP station for clarity.

Page 6, Line 7: IAEA (2014) this citation is listed in the references section as IAEA/WMO (2014). Please adapt.

Page 7, Line 16: based on instead of based off.

Page 8, Line 5: The authors should reconsider the terms "bahavioural" and "non-behavioural" for the model outputs of streamflow and stable isotope signature of streamflow. From my point of view those terms are not appropriate in this context. Reliable and non-reliable are terms coming to my mind here.

Page 8, Line 13: KGE. This abbreviation is introduced later (Line 23). Would be nice to have the explanation earlier.

Page 8, Line 30: Please mention for completeness that the other circa 52 % were sampled during the summer months.

Page 10, Line 11-18: Please explain clearer that you are talking about the average streamflow simulations of the three calibrations used in this paragraph. The reader will otherwise think you are talking about an average streamflow simulation (Line 12) of all model runs. Furthermore please precise which model you are talking about at the end of line 12 and beginning of line 13 ("The model also has...").

Page 10, Line 13: difficulty instead of difficultly

Page 10, Line 20-27: Please explain shortly why you have compared REMOiso vs. static and KNP43 vs. static for calculating the Kendall's tau coefficient.

Page 10, Line 29: Please revise the title of section 4.3 to Modelling delta oxygen-18 in streamflow).

Page 11, Line 14-16: Please check the literature and provide a reference here.

Page 11, Line 16: functions or function(s)?

Page 11, Line 21+22: Please provide some values (and percentages related to total annual precipitation) from mean annual precipitation for the mentioned periods (summer and fall, winter and spring).

Page 14, Lines 29-31: This sentence is a bit confusing. Please revise.

Page 14, Line 31: isoWATFLOOD or WATFLOOD?

Page 15, Line 4: isoWATFLOOD or WATFLOOD?

Page 15, Line 10: isoWATFLOOD or WATFLOOD?

Page 16, Line 13: kpn or KPN43?

Please check the citations carefully. Pietroniro et al. (1996) and Töyra et al. (1997) are listed in the references section (Page 18, Line 46 and Page 19, Line 34) but appear not in the manuscript itself.

In general, I liked the style and the coloring of the figures. However, figure 2 and 3 are a bit unclear. It is a real asset to show the uncertainty bounds of the different calibrations. The authors should rethink the presentation of this data, especially the streamflow results (panel b). Furthermore I would suggest indicating periods with snowfall and rainfall, if possible. At this point it would also make sense to combine the two times-series (static-rainfall and static-snowfall) to one static-precipitation input time-serie.

Figure 6: Are here shown the mean or median values (circle symbols)?

Figure 7: Please refer to Table 6 (were the parameters are explained) in the figure caption. The order of the table numbering in the text is sometimes were confused

(Page 4, Line 4: Table 1; Page 4, Line 32: Table 4, for example). Please order them correctly.

In general, I suggest reducing the amount of tables. Table 3 for example is not needed. The applied average correction values (and the range) can be mentioned in the text.

Table 5 is also unnecessary. You can mention the values in the text. However, it would be very relevant to explain in more detail how these values were selected.

Table 8 is also unnecessary from my point of view.

References: Dietermann, N. and Weiler, M.: Spatial distribution of stable water isotopes in alpine snow cover, Hydrology and Earth System Sciences, 17, 2657-2668, doi:10.5194/hess-17-2657-2013, 2013.

Taylor, S., Feng, X., Kirchner, J. W., Osterhuber, R., Klaue, B., and Renshaw, C. E.: Isotopic evolution of a seasonal snowpack and its melt, Water Resources Research, 37, 759-769, doi:10.1029/2000WR900341, 2001.

Taylor, S., Feng, X., Williams, M., and McNamara, J.: How isotopic fractionation of snowmelt affects hydrograph separation, Hydrological Processes, 16, 3683-3690, doi:10.1002/hyp.1232, 2002.

Unnikrishna, P. V., McDonnell, J. J., and Kendall, C.: Isotope variations in a Sierra Nevada snowpack and their relation to meltwater, Journal of Hydrology, 260, 38-57, 2002.

Zhou, S., Nakawo, M., Hashimoto, S., and Sakai, A.: The effect of refreezing on the isotopic composition of melting snowpack, Hydrological Processes, 22, 873-882, doi:10.1002/hyp.6662, 2008.

---

## Referee Comment (RC2) · C. Birkel (Referee) · 8 Dec 2016

The manuscript "Examining the impacts of estimated precipitation isotope ($\delta$18O) inputs on distributed tracer-aided hydrological modelling, Hydrol. Earth Syst. Sci. Discuss., doi:10.5194/hess-2016-539," by Delavau et al. currently under discussion in HESS highlights the importance of the input function and temporal resolution on tracer-aided modelling particularly in remote and data scarce catchments. The evaluation of large scale, spatially-distributed and climate model based isotope products as an alternative or complementary method to ground-based measurements could potentially become a feasible and widely used approach for tracer studies in areas with difficult access and monitoring constraints. I consider this as a novel contribution to the existing

literature. The paper is well-written and logically structured. It clearly demonstrates the impact of different isotope input functions on the coupled model and how this analysis contributes to constraining the model uncertainty particularly the internal functioning and how the model generates flows, mixing and the simulated water partitioning. Having said that, I think that the paper could be edited towards more clearly conveying the key points in terms of more generalizable results going beyond the Canadian context and the presented isoWATFLOOD model. I will detail my suggestions further below. Nevertheless, I am convinced that this paper will likely attract a lot of attention across a wide range of readers and beyond the hydrology community. Therefore, I am pleased to recommend this contribution for publication in HESS with some revisions.

All the best, Christian Birkel

Specific comments: My main point would be that the paper is in parts very much focussed on the particularities of the study site and also the presented model characteristics. However, the results and potential impact of this paper go in my opinion beyond this case study and this could be better emphasized to maximize impact particularly in the hydrological modeller community. I therefore, suggest the following:

- Title and Abstract: You could consider substituting the term "estimated" with e.g. "precipitation isotope product" throughout the manuscript to emphasize the different origins of the input functions. From Line 17 in the abstract, I suggest to revise these sentences, as they do not really reflect the key findings. For example, the statement that the model is only as good as its input function is rather trivial and could be changed to some more specific statement such as which temporal resolution is needed (hourly, daily, weekly. . .) to adequately simulate stream isotope signatures and which product is the best? I also suggest to more specifically mention that the coupled simulation of flow and isotopes actually allowed you to constrain the simulations towards a better internal representation of the dominating processes. - 2.2, Line 21:. . .is used "to" spatially distribute. . . - Page 7, Line 16: . . .based "on"? - Page 9, Line 14: Would it be feasible to test this for one model configuration and run it over let's say 100K iterations to be

able to check for differences compared to 30K runs? - Results and discussion: The results could be better linked to the wider literature. E.g. why not include the mean monthly precipitation isoscapes from Bowen and Revenaugh (2003) as a means of evaluation? I am missing a more concise attempt to generalize the results concerning model uncertainty and the value of tracer data in hydrological modelling. - Page 10, Line 1: How is the static approach with a single annual isotope value able to capture seasonal variability? - Conclusions and recommendations: I suggest to summarize the key points and present them in a numbered order. I also think it would be better to present the outlook as a separate section. - Would it be possible to include gridded maps of the different mean annual (and seasonal min/max) isotope products over the study area in relation to the observed data for comparison purposes?

---

## Author Comment (AC1) · 14 Jan 2017

**Table S1: Allowable parameter ranges and resulting averaged parameter characteristics for behavioural simulations retained from the three model calibrations (KPN43, REMOiso, and static inputs). Parameters are summarized as: median (minimum, maximum).**

| Parameter | Allowable Range | KPN43 | REMOiso | static |
|---|---|---|---|---|
| **Jean Marie Sub-basin** | | | | |
| **flz** | $[1 \times 10^{-7}, 1 \times 10^{-4}]$ | $8.60 \times 10^{-6}$ ($1.49 \times 10^{-7}$, $9.32 \times 10^{-5}$) | $8.60 \times 10^{-6}$ ($2.03 \times 10^{-7}$, $9.86 \times 10^{-5}$) | $8.60 \times 10^{-6}$ ($2.03 \times 10^{-7}$, $9.86 \times 10^{-5}$) |
| **pwr** | [1, 4] | 3.05 (1.33, 4.00) | 3.38 (1.12, 4.00) | 3.39 (1.12, 3.96) |
| **theta** | [0.1, 1.0] | 0.60 (0.13, 0.96) | 0.61 (0.13, 1.0) | 0.60 (0.13, 0.98) |
| **kcond** | [0.1, 1.5] | 0.79 (0.17, 1.46) | 0.68 (0.17, 1.46) | 0.86 (0.17, 1.46) |
| **Blackstone Sub-basin** | | | | |
| **flz** | $[1 \times 10^{-7}, 1 \times 10^{-4}]$ | $2.30 \times 10^{-5}$ ($4.19 \times 10^{-7}$, $9.41 \times 10^{-5}$) | $4.91 \times 10^{-5}$ ($5.36 \times 10^{-7}$, $9.95 \times 10^{-5}$) | $1.67 \times 10^{-5}$ ($3.02 \times 10^{-7}$, $9.86 \times 10^{-5}$) |
| **pwr** | [1, 4] | 3.34 (1.82, 3.96) | 3.21 (1.35, 3.96) | 3.57 (2.28, 3.91) |
| **theta** | [0.1, 1.0] | 0.55 (0.11, 1.0) | 0.60 (0.13, 1.0) | 0.52 (0.24, 1.0) |
| **kcond** | [0.1, 1.5] | 0.69 (0.11, 1.41) | 0.80 (0.21, 1.48) | 0.69 (0.17, 1.49) |
| **Land Cover Weighted-Average Parameter results: median (minimum, maximum)** | | | | |
| **fratio** | [0.1, 2.5] | 0.70 (0.12, 2.22) | 0.70 (0.15, 2.16) | 0.80 (0.12, 2.23) |
| **ak** | [1, 50] | 21.6 (2.1, 47.6) | 25.8 (2.4, 47.5) | 23.6 (1.2, 46.9) |
| **akfs** | [0.005, 2] | 0.212 (.006, 1.878) | 0.059 (0.006, 1.724) | 0.203 (0.006, 1.850) |
| **rec** | [0.05, 1] | 0.47 (0.08, 0.90) | 0.46 (0.08, 0.88) | 0.43 (0.09, 0.90) |
| **retn** | [10, 200] | 122 (18, 189) | 119 (23, 181) | 114 (20, 186) |
| **ak2** | [.001, 0.2] | 0.013 (0.001, 0.188) | 0.008 (0.001, 0.172) | 0.021 (0.001, 0.184) |
| **fm** | [0.075, 0.2] | 0.117 (0.076, 0.189) | 0.119 (0.078, 0.190) | 0.112 (0.076, 0.189) |
| **base** | [-3.5, 3.5] | -0.20 (-3.28, 3.14) | 0.34 (-2.81, 3.13) | -0.35 (-3.11, 2.85) |
| **sub** | [0.1, 1.1] | 0.53 (0.11, 1.05) | 0.43 (0.13, 1.05) | 0.50 (0.11, 1.05) |

[Figure]

**Figure S-1: Spatial distribution of precipitation isotope product $\delta^{18}O_{ppt}$ input to isoWATFLOOD (10 km resolution) for static (first column), REMOiso (second column) and KPN43 (third column). $\delta^{18}O_{ppt}$ was flux-weighted using gridded WATFLOOD precipitation input and averaged daily by season over the study period (1997-1999): DJF (a to c), MAM (d to f), JJA (g to i) and SON (j to l).**

---

## Author Comment (AC2) · 14 Jan 2017

We sincerely thank both referees for their thorough reviews and most constructive comments on our manuscript (Reference HESS-2016-539). We fully recognize and appreciate the reviewers' efforts in providing these informative reports on our research and their insights have led to an improved interpretation of our results. We have therefore taken into full consideration all of these comments and have prepared responses to these as well as information on how the paper was revised following the referees' suggestions. Our responses and edits to the paper are provided below in bold following the individual comments requiring action from reviewer 2, Dr. Christian Birkel.

Please do not hesitate to contact us if any of this information is not clear.

With kind regards,
Tricia Stadnyk (on behalf of all co-authors)

Referee 2 The manuscript "Examining the impacts of estimated precipitation isotope (18O) inputs on distributed tracer-aided hydrological modelling, Hydrol. Earth Syst. Sci. Discuss., doi:10.5194/hess-2016-539," by Delavau et al. currently under discussion in HESS highlights the importance of the input function and temporal resolution on tracer-aided modelling particularly in remote and data scarce catchments. The evaluation of large scale, spatially-distributed and climate model based isotope products as an alternative or complementary method to ground-based measurements could potentially become a feasible and widely used approach for tracer studies in areas with difficult access and monitoring constraints. I consider this as a novel contribution to the existing literature. The paper is well-written and logically structured. It clearly demonstrates the impact of different isotope input functions on the coupled model and how this analysis contributes to constraining the model uncertainty particularly the internal functioning and how the model generates flows, mixing and the simulated water partitioning. Having said that, I think that the paper could be edited towards more clearly conveying the key points in terms of more generalizable results going beyond the Canadian context and the presented isoWATFLOOD model. I will detail my suggestions further below. Nevertheless, I am convinced that this paper will likely attract a lot of attention across a wide range of readers and beyond the hydrology community.
**Thank you kindly for your summary and assessment of our paper, Dr. Birkel, and we agree that we are excited about the implications this manuscript and its comparison of isotope precipitation products may have on the isotope-enabled modelling world. We believe the changes you've suggested have greatly improved the quality of this manuscript.**

Specific comments: My main point would be that the paper is in parts very much focussed on the particularities of the study site and also the presented model characteristics. However, the results and potential impact of this paper go in my opinion beyond

this case study and this could be better emphasized to maximize impact particularly in the hydrological modeller community. I therefore, suggest the following:

**We also agree that the findings presented in this manuscript go beyond our specific application to the Fort Simpson region and are therefore more general and impactful than we have conveyed them. We have edited the manuscript in a way that conveys our findings in a more general sense, specifically with respect to a range of study sites (particularly those that have seasonality as this one), isotope-enabled models, and modelling applications. Thank you for this feedback.**

- Title and Abstract: You could consider substituting the term "estimated" with e.g. "precipitation isotope product" throughout the manuscript to emphasize the different origins of the input functions.

**We like this terminology and have adopted it for the revised title *Examining the impacts of precipitation isotope products ($\delta^{18}$O) on distributed tracer-aided hydrological modelling*, as well as throughout the paper. Thank you for the suggestion!**

From Line 17 in the abstract, I suggest to revise these sentences, as they do not really reflect the key findings. For example, the statement that the model is only as good as its input function is rather trivial and could be changed to some more specific statement such as which temporal resolution is needed (hourly, daily, weekly...) to adequately simulate stream isotope signatures and which product is the best?

**Thank you for this suggestion, and we also agree. We have reworded the abstract to instead state "We investigate the impact that choice of precipitation isotope product ($\delta^{18}$O$_{ppt}$) has on model simulations of streamflow, d18O of streamflow, and model parameterization in high-latitude, highly seasonal regions. We assess three precipitation isotope products (i.e., one new, two from the literature) of different spatial and temporal resolutions, and apply them as forcing to the isoWAT-FLOOD tracer-aided hydrological model in the Fort Simpson, NWT basin." And**

**perhaps more importantly, we have revised our discussion and conclusions to comment specifically on the impact that precipitation isotope product resolution has on model output. This has become one of our key take-home messages.**

I also suggest to more specifically mention that the coupled simulation of flow and isotopes actually allowed you to constrain the simulations towards a better internal representation of the dominating processes.
**We agree and have revised the last sentence in our abstract to state:** *Furthermore, the application of a tracer-aided model constrained simulations to achieve a better internal representation of watershed processes, reinforcing that a tracer-aided modelling approach assists with resolving hydrograph component contributions, and works towards diagnosing model equifinality.*

- 2.2, Line 21:. . .is used "to" spatially distribute. . .
**Corrected, thank you.**

- Page 7, Line 16:. . .based "on"?
**Corrected.**

- Page 9, Line 14: Would it be feasible to test this for one model configuration and run it over let's say 100K iterations to be able to check for differences compared to 30K runs?
**Feasible, absolutely. In the time we have for edits to be submitted for this manuscript – no (we estimate it would take minimum 1 month, perhaps longer). That said, we are in the process of doing 100k runs with (iso)WATFLOOD in another northern basin to look at parameter identifiability with and without the use of isotopes in model calibration and nearing the end of those runs. We are planning to submit this manuscript for peer review within the next couple of months, where we will more definitively tackle the issue of parameter identifiability. Though we think this is a critical issue, it is not the intended focus of this manuscript, but rather follow up work that we now (more clearly) describe in the**
new *Future Directions* section of this manuscript.

- Results and discussion: The results could be better linked to the wider literature. E.g. why not include the mean monthly precipitation isoscapes from Bowen and Revenaugh (2003) as a means of evaluation?

**This is an interesting suggestion, however, this would only further evaluate KPN43 and REMOiso products and not $\delta^{18}O_{sf}$ or other types of simulation output that are our intended focus. Bowen and Revenaugh's 2003 isoscapes are derived from long term average global models that did not include any CNIP data within their formulation, so we aren't convinced this would be a good dataset from which to further validate our REMOiso or KPN43 estimates of $\delta^{18}O_{ppt}$ over the Fort Simpson region. It should be pointed out that the KPN models have already been evaluated rigorously in Delavau et al., 2015. REMOiso could definitely use more validation in Canada, but that was already mentioned, we're not confident that the Bowen Revenaugh 2003 isoscape would help with this. We have listed this instead as future work, and it is not currently within the scope of this study to evaluate REMOiso outside of the Fort Simpson study area. That being said, we believe that a comparison (to Bowen Revenaugh's isoscape) may show that KPN43 is a better estimate of $\delta^{18}O_{ppt}$ in Canada than the other global models. On a cautionary note, however, this would be comparing oranges to apples because the time scale of the isoscapes would not be the same. And finally.... the static values were derived from actual observations, so there is no need to make a comparison to Bowen there.**

I am missing a more concise attempt to generalize the results concerning model uncertainty and the value of tracer data in hydrological modelling.

**We agree and have revised the discussion section of the manuscript – and conclusions – extensively to help draw these generalized results into take-home conclusions for the broader tracer-aided modelling community.**

- Page 10, Line 1: How is the static approach with a single annual isotope value able

to capture seasonal variability?

**So the static approach is actually two annual isotope values: one for rainfall and one for snowfall. Therefore, technically speaking, the static approach is capable of capturing some seasonality. This is a point we have much more clearly (and in more detail) described in the manuscript. The fact that the static input captures "sufficient seasonality" is likely more a function of our high-latitude study site than the value of a static input alone. Namely, in high-latitude environments, particularly Fort Simpson, there is no mid-winter freeze/thaw/melt – resulting in snowpack accumulation throughout the entire winter season and one significant freshet in late spring. Similarly, soils freeze up as does any soil moisture that may in other regions contribute to baseflow and/or streamflow throughout the winter. In high-latitude regions, seasonality is more binary than quarterly, therefore the two annual static inputs do a reasonable job of capturing the seasonality.**

- Conclusions and recommendations: I suggest to summarize the key points and present them in a numbered order. I also think it would be better to present the outlook as a separate section.

**We have taken your suggestion to mean a numbered summary of the key takehome messages, which we have better aligned with the objectives and numbered accordingly in the conclusions section. With regards to "outlook", we assumed you mean future work to be done with the modelling, and have added a "Future Directions" section to this manuscript.**

- Would it be possible to include gridded maps of the different mean annual (and seasonal min/max) isotope products over the study area in relation to the observed data for comparison purposes?

**Thank you for this suggestion. Though we don't feel another figure is warranted in the manuscript, we see the value in these figures and the presentation of our precipitation isotope products for the modelling community and have decided to add it as a supplement to our manuscript (Figure S-1). To generate the spatially**

distributed precipitation isotope products maps, daily isotope in precipitation input used to drive the distributed tracer-aided model was averaged daily across each season (DJF, MAM, JJA, SON) for each source (static, REMOiso, KPN43). Maps were generated using the model grid (10k) and entire modelling domain (includes both Jean-Marie and Blackstone), and isotope compositions were flux-weighted using daily distributed (10 k) precipitation input to WATFLOOD (interpolated Environment Canada station observation, housed in WATFLOODs radcl .r2c files; Kouwen 2014). The resultant maps indicate clear differences in spatial variability among the inputs. Static – not surprisingly – is spatially constant (as it should be!), but seasonally variant resulting from the mixture of rain and snowfall events on the shoulder seasons (MAM and SON). REMOiso has less variability than the KPN43 input, resulting from REMOiso's 55 km grid resolution (i.e., approx.. 5 of the isoWATFLOOD grids shown on our Figure) which would act to smooth topographical and land cover differences in part driving changes in isotopic composition. We've added a brief discussion to the paper and reference to Figure S-1. For your interest and review – we also generated a figure (not included in the manuscript) averaged across the entire study period (1997-1999) for each model input (Figure 1). This confirms the enhanced spatial variability from the KPN43 model, followed by REMOiso (derived from a 55km RCM), and the spatially constant Static input. Because of the high-latitude of the study region, the static input shows that snowfall prevails over rainfall for this site (in terms of isotopic composition), and that the 3-year annual average is more depleted than the temporally (and spatially) variable inputs. KPN43 variability is enhanced in the 3 year average because it is more consistent from grid-to-grid in each year (driven by the KPN43 regionalization) than REMOiso, which would vary temporally and spatially daily and from year to year. We could not generate an observed isotope in precipitation map because we did not have enough observed data to so.

[Figure]

[Figure]

**Fig. 1.** Spatial distribution of precipitation isotope products averaged across the entire study period (1997-1999)

---

## Author Comment (AC3) · 14 Jan 2017

We sincerely thank both referees for their thorough reviews and most constructive comments on our manuscript (Reference HESS-2016-539). We fully recognize and appreciate the reviewers' efforts in providing these informative reports on our research and their insights have led to an improved interpretation of our results. We have therefore taken into full consideration all of these comments and have prepared responses to these as well as information on how the paper was revised following the referees' suggestions. Our responses and edits to the paper are provided below in bold following the individual comments requiring action from reviewer RC1.

Please do not hesitate to contact us if any of this information is not clear.

[Figure]

With kind regards,
Tricia Stadnyk (on behalf of co-authors)

Referee 1:
General comments: The authors present an interesting case study about using differing precipitation stable isotope input datasets for distributed hydrological modeling in the Northwest Territories of Canada. Based on three different precipitation stable isotope datasets, three different calibrations of the model isoWATFLOOD were identified based on a Monte Carlo random sampling approach. The results show that modeled streamflow was relatively similar for each of the three used stable isotope input datasets. Whereas the differences in the modeled stable isotope signature in streamflow and the internal apportionment were much more pronounced. However, the study is lacking some important and critical explanation of the presented model outputs. Please find a detailed description of specific comments and technical notes below. The focus of the presented study is in the scope of HESS. Further more I will highlight at this point that the paper is very well written, understandable and the sections are mostly very well structured. However, based on my review below I recommend major revisions prior to a publication in HESS. **Thank you kindly for this summary of our paper. Indeed, we agree based on the reviewers assessment that the discussion can be enhanced with respect to model outputs – we thank you for your detailed assessment and guidance provided. We believe the changes you've suggested have greatly improved the quality of this manuscript.**

Specific comments:
The authors should rethink the use of the word "estimated" in the title as well as throughout the whole manuscript. It suggests that the input data was generated specifically for the presented study. It should be clear that (2 of 3) available precipitation isotope product were used to the study. Which is actually an asset for the study and with respect to future studies in other basins. **We agree completely, and this was also suggested by the second reviewer too. We have changed the title to *Examining***

[Figure]

*the impacts of precipitation isotope products ($\delta^{18}O$) on distributed tracer-aided hydrological modelling* **and revised the use of the word 'estimated' (with respect to $\delta^{18}O_{ppt}$ inputs) throughout the manuscript to** *precipitation isotope products,* **as appropriate.**

The first sentence of the abstract is "…increasingly popular tools as they have documented utility in constraining model parameter space during calibration, reducing model uncertainty, and assisting with selection of appropriate model structures.". However, there is no evidence for that statement. Please include additional information to the introduction section or revise the first sentence of the abstract. **We have subsequently revised the abstract significantly, and agree that it has yet to be proven that the parameter space is constrained by such tools. We are currently conducting such a study within our group (though it has yet to be published), and do have internal evidence that this is the case. That said, we have rephrased this sentence as:** *Tracer-aided hydrological models are becoming increasingly popular tools as they assist with process understanding and source separation, which aides in model calibration and the diagnosis of model uncertainty (Tetzlaff et al. 2015; Klaus McDonnell, 2013).*

The authors highlight the importance of snowmelt in the study region. The stable isotope signature of the snow pack and its melt water is a very challenging topic. Please handle this point very carefully in your publication. On page 5, Line 17 for example you mention that the default method for oxygen-18 input is annual average rainfall and snowfall. In your static approach, however, you used average measurements of rainfall and snowpack from the GEWEX campaign. Please provide the values of snow pack stable isotope signature in figure 5 by the way. Especially during the ablation season the isotopic evolution of the snowpack progresses due to percolating rain water and fractionation caused by processes like melting and sublimation (Zhou et al., 2008; Unnikrishna et al., 2002; Dietermann and Weiler, 2013; Lee et al., 2010). This leads to

an increase of heavy isotopes in melt water throughout the freshet period (Taylor et al., 2001, 2002; Unnikrishna et al., 2002). Which is correctly represented by the shown model results. Taylor et al. (2001 and 2002) point out that for hydrological applications (in their case isotope based hydrograph separation) a correct representation of the snow pack melt water is absolutely crucial.

**Thank you for your insight, and we couldn't agree more that the isotopic signature of a snowpack and its evolution in snow melt are very challenging processes. We have been studying this topic for more than five years now, in part through an IAEA coordinated research project experimenting with methods to collect isotopes in snowmelt (Penna et al., 2014), and looking at seasonal changes globally between snowpack composition and snowmelt.**

**This being said, in this manuscript we need to be diligent in how we handle the topic since we did not collect the isotopes in snow data, and there is no specific legacy of how or where it was collected from (i.e., from what part of the snow-pack, averaged depth dependent or composite samples, and unknown spatial variability of the samples). This is one of the reasons why we chose not to in-clude the snowpack compositions on Figure 5 originally. We have since revised the figure and added the snowpack data and also included a cautionary note to readers highlighting there is uncertainty surrounding these measurements. The revised Figure 5 is included in this response (Fig. 1).**

**For the modelling, as a static input our model would preferably use average annual inputs of rainfall and snowfall. Rainfall values were, as reviewer 1 notes, obtained from the GEWEX campaign. However, there was no data on snowfall composition available – only snowpack compositions - therefore (as in several of other data limited, high latitude tracer-aided modelling studies: Stadnyk et al. 2013; Smith et al., 2015; Smith et al., 2016; Holmes 2016), we assume (as model input) that the average annual composition of snowfall is approximately**

**equal to that of the snowpack samples from the GEWEX campaign. From our own experiments, we know this is not always the case (typically true only in the short term immediately after a snowfall). With no other snowfall composition data, however, this is an assumption we have been required to make. We have clarified our assumption in the manuscript. We would also like to point out that due to the high latitude of this and several other of our sites, freeze-thaw cycles common in snowpacks are in fact rare in high-latitude (northern Canadian) snowpack where temperatures remain significantly below freezing for the entire winter season – as was found when we compared our (southern in comparison to this field site) Winnipeg, MB, Canada site to the other IAEA study sites included in Penna et al., 2014.**

**Lastly, we absolutely agree the representation of snowfall, snowpack and snow melt compositions in modelling (particularly high-latitude, seasonal regions) is absolutely crucial, which is why our group is putting extensive resources into re-solving some of the uncertainty surrounding these processes and the evolution of isotopic compositions through these processes. Thank you for your feedback, and for reaffirming the importance of this issue!**

REMOiso is a distributed dataset and the precipitation amounts are also available spatially distributed over the study area. Why was the precipitation amount weighting only conducted at one location and not spatially distributed? **We are not entirely sure of what you mean by this question, but will attempt to answer it as best we can within the context of what we did in this study. The only precipitation amount-weighting for REMOiso was done to determine the bias correction at Snare Rapids. There was no need to do this spatially for this purpose as we are comparing CNIP observations (at a point) directly to REMOiso output at a single location corresponding to the location of the CNIP observation station. Now, if what you are getting at is why did we not precipitation amount weight**

**REMOiso using REMO-derived precipitation for use in this study, then our answer is as follows. We averaged the four 6-hourly REMOiso values (at each grid) to arrive at daily compositions that were directly read into the model as input on a per-grid basis (i.e., no amount-weighting involved – same as for the static and KPN inputs). Based on some (unpublished) analyses we did for a study of the Mackenzie River Basin (i.e., using the same REMOiso model output), we don't trust the quality of sub-daily REMO precipitation to the point where we would use (sub-daily) precipitation to amount weight REMOiso $\delta^{18}O_{ppt}$ (i.e., in the same way that we are skeptical of sub-daily REMOiso $^{18}O$ compositions, which is part of the reason that we decided to average output daily to avoid unrealistic variation). If we decided to amount weight, we couldn't use actual observations to amount weight 6-hourly to daily as we only have daily precipitation from Fort Simpson Airport and at various grid locations from the ANUSPLIN product. We hope we have addressed your question and your concerns.**

The authors mention that "several changes and improvements" (Page 7, Line 16) were carried out in the model version used for the study. In the following only one modification (proportion of bog an fen split) is mentioned. Are there any other modifications? If so, please mention them here.

**This was poorly worded on our part. What we meant to say was that the model (isoWATFLOOD) has undergone "several changes and improvements" since it was last published in a study back in 2013 (Stadnyk et al., 2013). These changes and improvements were independent of the current study, and all toward continual improvement of internal dynamics and the model output. We have revised the wording in our manuscript to clarify:** *The model used in this study (isoWATLOOD) is based on the version used by Stadnyk et al. (2013), noting that a different version of isoWATFLOOD and the Fort Simpson watershed model were used here that incorporates various model improvements made since 2013, independent of this study*

The first two paragraphs of section 4 (Results and discussion) should definitively be revised. There is a lot of content that can be mentioned later in the conclusions section (the last sentence on Line 12-14 for example). **We have significantly revised the results discussion using the guidance of your questions below to help highlight specific findings related to our key objectives and take-home messages. We have also moved the sentence you reference above to the conclusions.**

In section 4.2 (Modelling streamflow) please explain the model results as well as the observed streamflow in much more detail. The three different inputs (and three different calibrations) provide very similar results for the simulated streamflow (Page 15, Lines 5-8). Those results should be discussed in more detail. **Thank you for your suggestions, we have revised the discussion to include more specific, in-depth discussion of the simulated streamflow resulting from the three types of precipitation isotope product. And yes, all three precipitation isotope products (three different calibrations) result in almost exactly the same streamflow simulation (i.e., statistically the same according to the Kendall's tau test applied in the paper). This is the core definition of equifinality, illustrated here in this study! Despite there being significant differences in the parameters, the net result of the simulation remains almost identical (driven by the requirement for the model to meet specific efficiency criteria). This can happen by changing how and where water is stored internally in the model – greatly affecting the transit time of water through the model and into the stream – but ultimately not impacting the total flow simulated by the model (because various internal processes trade-off in their respective contributions). Again, this was highlighted in this study by the fact that upon closer examination of the model simulations, internal apportionment of water was significantly altered from one input (calibration) to the next, particularly when comparing REMOiso to the KPN and static calibrations. This the very essence of our study! Your specific questions regarding this section of the discussion are answered below.**

Is there really no discharge in winter (Figure 2 and 3)? **We assume you are referring to Figures 3 & 4 (not 2). And no, observed streamflow does not go zero, but rather becomes very small relative to peak flows: minimum in Jean-Marie from 1997-1999 of 0.194 m$^3$/s, or 0.5 percent of the maximum streamflow, 35 m$^3$/s during this same period; and a minimum of 0.043 m$^3$/s in Blackstone relative to a maximum flow of 109 m$^3$/s, so less than 0.04 percent of the peak flow. Ice-on winter low flows in high latitude basins such as this commonly reduce significantly and become near zero due to the long, sustained period frozen ground/soils, lack of mid-winter thaw/melt periods, and accumulation of solid precipitation.**

**We considered providing panel b on Fig 3 & 4 in log-scale to emphasize that there are in fact low-flow values, but this greatly diminished peak flow analysis and peak flow uncertainty which was a key point in our study. E.g., Figure 3 panel b in log-scale, which is not included in the revised manuscript but is provided in this response for your reference (Fig. 2). We have added some text regarding low flow, ice-on streamflow values to the manuscript though.**

What are the influences of groundwater on the hydrology of the region? The same holds for section 4.3. Explain the results in more details. **Given the region resides within the discontinuous to semi-permafrost region of Canada, the influence of sub-surface contributions to runoff would be sporadic and is difficult to define (as several studies in the region have shown, Connon et al., 2015). We would argue that groundwater is not as influential as the bog complexes (or bog cascades as Connon et al., 2015 defined them), which depending on wetness levels, interconnect and disconnect seasonally and inter-annually. The model we use in this study (isoWATFLOOD) has the capability to raise/lower wetland water table levels, connecting and/or disconnecting with channel runoff, which is a reasonable analogy to this complex interaction.**

There is especially the time of the spring freshet that needs much more carefully discussed. **We have incorporated an analysis of the results during spring freshet into our discussion.**

The model results show a sharp drop of streamflow stable isotope signature, while the observed values are getting more and more enriched at that time. This completely opposed development may be related that the contribution of snowmelt water to total streamflow during that time is too high (or the signature of the snow melt signal is wrong, please remind here my suggestions above) and the contribution of baseflow too low. **We assume you are referring to the freshet period in 1998. Note that we did not have continuously observed isotopes in streamflow during the peak freshet (i.e., high flow sampling is not always feasible), and as a result there are some missing observations during this time of year (mostly in 1999), despite this being our most frequent period of sampling overall (relative to other seasons). Moreover, as we've explained the model assumes snowfall composition to be equal to snowpack composition, and then can apply a constant offset or fractionation from snowpack composition/accumulation to snowmelt. In this study, that offset was set =0 given the lack of snowpack to snowmelt observations from which to calibrate to. Therefore, it is most likely that, in this year, the assumed fractionation from pack to melt water was wrong and not well defined. Again – without observed data to compare to, it is impossible for us to adjust this factor to improve results; however, adding a snowmelt dynamics module to the model would be a great asset, one which has been recognized by our group and that we are working toward. It is likely that the timing of the simulated snowmelt contribution to runoff from the model resulted in what appears to be an overlap between the most depleted simulated isotopic composition of streamflow with observational data that is enriching (i.e., which is in fact post-**

**freshet in 1998 and 1999 due to the presence of a strong El Nino event, noted by St. Amour et al 2005, and occurring due to evaporative fractionation). Similarly, in 1999, note the gap between ice-on observed compositions of streamflow and the enriching values (i.e., again post-freshet and occurring due to evaporative enrichment) relative to the most depleted simulated isotopic composition of streamflow, occurring around the same time due to snowmelt-driven runoff (i.e., and a later-than-observed snowmelt period in the model, more clearly distinguishable in 1999, Fig 3, panel b). These differences are likely occurring as a result of differential warming (rate and onset) caused by the El Nino event in 1998 and somewhat in 1999 that results in contrasting behaviours (from 1997, but also between observations and our calibrated model). Regarding the baseflow or groundwater composition comment you had, please see our response below (next comment/response).**

The contributions of baseflow (groundwater) to total streamflow during the post-freshet are especially for Jean Marie River much lower in the present model study compared to the results of St Amour et al. (2005). Furthermore, please provide the stable isotope signature of groundwater. And compare those observed values with the values generated by the models in the groundwater routine after the spin-up period. **As Stadnyk et al. (2005) and Stadnyk-Falcone (2008) pointed out, contributions of "groundwater" from the model (isoWATFLOOD) cannot be directly compared to those derived by St. Amour et al. (2005) owing to the definition of what groundwater is considered in the two modelling methodologies. In St. Amour et al (2005), a mixing model is used that separates old and new water contributions over time – which means that groundwater is defined as old water, or that is water that is existing pre-event. Whereas using WATFLOOD (Stadnyk et al. 2005) or isoWATFLOOD (Stadnyk-Falcone, 2008) to perform hydrograph separation in the same region, lower contributions of groundwater are derived by the (iso)WATFLOOD model since the model separates soil water (upper zone storage) from baseflow or groundwater (lower zone storage) and wetland**

storage – all of which would constitute 'old' (pre-event) water using traditional two-component mixing models. Regarding groundwater isotopic composition: if groundwater was sampled, we do not have the data. It was our understanding (from speaking with Natalie St. Amour) that her 2005 paper used ice-on low flow to define baseflow or groundwater contribution (Table IV, St. Amour et al., 2005). Her paper suggests groundwater compositions are -20.5+/- 0.8 per mille (in 1998) and -20.4 +/- 1.0 per mille (in 1999). Uncertainty is due to averaging across all the five Fort Simpson basins. Modelled groundwater compositions in isoWATFLOOD were found to be 40-70 percent and 60-70 percent for Jean Marie and Blackstone, respectively during the post-freshet (JJASON) period.

Please check the manuscript for repetitive information. The sentence on Page 13, 34+35 for example appears almost identical on the next page again (Page 14, Lines 20-22). This would be an excellent take-home sentence for the conclusions section by the way. **Thank you for pointing this out. We have re-read the manuscript and removed any apparent redundancies, particularly the ones you've pointed out to us. We have moved the sentence you highlighted to the conclusions section.**

In general, I am missing some distinct conclusions in the conclusions section of the submitted manuscript. There are a lot of recommendations and speculations but no clear take-home messages. **Agreed. In re-reading the manuscript, we too realized that we can write better conclusions that highlight the take-home messages this manuscript presents. Our conclusions now start with the following numbered take-home conclusions, which are further elaborated on in the conclusions section; specifically,** *that choice of precipitation isotope product: 1. Does not impact simulation of total streamflow; 2) Impacts model parameterization, and therefore modelling uncertainty; 3) Impacts internal apportionment of water in the model (through model parameterization), impacting resultant hydrograph separation -*

*and therefore simulated transit times of water; and 4) impacted $\delta^{18}O_{sf}$ most significantly when event composition differed significantly from streamflow composition (e.g., snowmelt and large rainfall events).*

**Also elaborated on in the conclusions now is the take-home message that precipitation isotope products of higher resolution (e.g., REMOiso, daily resolution) better capture event-specific compositions that, when significantly different from $\delta^{18}O_{sf}$, tend to cause significant deviations from seasonal and semi-annual (i.e., static) inputs. Though we cannot verify the correctness of the higher resolution product (REMOiso) in this study due to monthly observed precipitation, it is clear that temporal resolution plays a significant role in model parameterization and resulting hydrograph separations. We have also added a separate *Future Directions* section (based on Reviewer 2 feedback) that is comprised of the future work discussion from our original conclusions.**

Technical notes:
Page 1, Line 18: "...to capture both the variability and seasonality". There it would be better to write "spatial variability and seasonality" or "spatial and temporal variability", since the seasonality is also a variability (temporal). **We have made this correction.**
Page 1, Line 31: (e.g. Beven and Binley, 1992; Kirchner. ...) **Correction made.**
Page 3, Line 22 and Line 29: Please provide size and elevation characteristics of the basins here. **We have added this information.**
Page 3, Line 27: "...is selected based ON data availability." **Correction made.**
Page 4, Line 20: The study region is not a high elevation region. Please mention correctly why the approach is suitable for the study region. **From another project our research group is working on, a detailed analysis of ANUSPLIN's suitability for high latitude, Boreal regions (i.e., specifically the Nelson River) was done by a PhD student (Rajtantra Lilhare) and presented recently in a poster at AGU (Lilhare, 2016). In this study, both the seasonality and amount of precipitation from ANUSPLIN were found to match well with observations from three nearby (within**

the Nelson River watershed) Environment Canada meteorological station obser-vations. Simultaneously, we have been involved in an assessment of precipitation datasets and reanalysis products across the Canadian Prairies and Boreal region for the purposes of hydrological modelling applications. ANUSPLIN was included in this comparison, where data products were evaluated against independent station data (not used in the derivation of each product). A manuscript summarizing this comparison is currently in preparation by Dr. Bruce Davison, who found that ANUSPLIN scored well in terms of accuracy (relative to station observations), but showed some bias over the long-term. Based on our knowledge of ANUSPLIN for our study area, we believe that it is adequate to describe daily precipitation over the short term, but this decision would need to be reconsidered should the study length be extended.

Page 4, Line 21: "...is used TO spatially..." **Correction made.**

Page 5, Line 9: Why are they not adequate for model forcing? This is the input data used and referred STATIC in the study, right? Please revise this sentence **Our apologies. We have revised this sentence to instead state:** ...*their spatial and temporal resolutions are not preferred for tracer-aided hydrologic model forcing due the observations being uniform in space, and their poor temporal resolution.*

Page 5, Line 12: "such that" appears twice. **Corrected.**

Page 5, Line 24: KP43 instead of KPN43. **Corrected – thank you for noticing this!**

Page 6, Line 4: From my point of view the section 2.4.1 is a description of methods and should therefore be moved in the appropriate section. **We agree and have moved this section to a new section in study methods.**

Page 6, Line 16: Please mention that Snare Rapids is a CNIP station for clarity. **We have added this information and clarified.**

Page 6, Line 7: IAEA (2014) this citation is listed in the references section as IAEA/WMO (2014). Please adapt. **This has been corrected.**

Page 7, Line 16: based on instead of based off. **Corrected.**

Page 8, Line 5: The authors should reconsider the terms "behavioural" and "nonbehavioural" for the model outputs of streamflow and stable isotope signature of streamflow. From my point of view those terms are not appropriate in this context. Reliable and non-reliable are terms coming to my mind here.

**These terms are not our own and are taken from the modelling literature referring to whether or not a simulation meets the threshold criteria value (based on efficiency criteria for each study – and defined here as a combination of percent Dv, log(percent Dv), NSE, KGE, and RMSE) to remain included in the final analysis. The term behavioural refers to the fact that the simulation (and therefore parameters driving the simulation) are adequately describing the behaviour of the environmental system (i.e., hydrological response). Since this terminology is historically well defined in the model calibration and equifinality literature (e.g., Tolson Shoemaker, 2008; Beven Freer, 2001; Zak Beven, 1999; Beven Binley, 1992, . . .), we would prefer not to deviate from the accepted terminology. Moreover, we don't believe the term reliable captures what we are doing here. Multiple simulations can all have the same statistical likelihood, therefore all reliably predict a given result (statistical likelihood, or efficacy criteria). But some may do so with parameter values that are unrealistic and not representative of the environmental system (i.e., non-behavioural).**

Page 8, Line 13: KGE. This abbreviation is introduced later (Line 23). Would be nice to have the explanation earlier.

**Though we see your point, it would clutter the step-by-step methodology and we feel it would be out of place to put the statistic description further up. We have instead noted that the statistic is described below for readers who are unfamiliar with it.**

Page 8, Line 30: Please mention for completeness that the other circa 52 **We have added this for clarification.**

Page 10, Line 11-18: Please explain clearer that you are talking about the average streamflow simulations of the three calibrations used in this paragraph. The reader will otherwise think you are talking about an average streamflow simulation (Line 12) of all

model runs. Further more please precise which model you are talking about at the end of line 12 and beginning of line 13 ("The model also has..."). **Thank you for pointing this out. We agree and have revised this portion of the discussion to be much more specific to which runs we are referring (i.e., all models, the range and/or mean of the models, or a specific model derived from a particular $\delta^{18}O_{ppt}$ input).**
Page 10, Line 13: difficulty instead of difficultly **Corrected – again, impressive that you noticed this! Many thanks.**
Page 10, Line 20-27: Please explain shortly why you have compared REMOiso vs. static and KNP43 vs. static for calculating the Kendall's tau coefficient. **We in fact calculated Tau for all possible comparisons (ie. KPN vs. REMOiso, KPN vs. static, REMOiso vs. static) for both basins, but did not report all values in the manuscript, but instead reported only the range of the values by selecting these specific pairings. Moreover, Since static represents $\delta^{18}O_{ppt}$ observations (annual average), by comparing REMOiso and KPN43 directly to static, we are in essence comparing them to simulations derived from mean annual $\delta^{18}O_{ppt}$ observations.**
Page 10, Line 29: Please revise the title of section 4.3 to Modelling delta oxygen-18 in streamflow). **Done.**
Page 11, Line 14-16: Please check the literature and provide a reference here. **We have provided the following reference where the authors looked a comparison of a decomposition of the NSE and KGE stats: Kling, H.V., and H. Gupta 2009.**
Page 11, Line 16: functions or function(s)? **Functions. We have corrected this.**
Page 11, Line 21+22: Please provide some values (and percentages related to total annual precipitation) from mean annual precipitation for the mentioned periods (summer and fall, winter and spring). **We are a tabular summary that includes a percentage breakdown for seasonal (summer/fall, or JJASON and winter/spring, or DJFMAM) snowfall and rainfall during our study period (1997-1999) in this response (Fig. 3). In comparison to the long-term climate normal (1981-2010) at Fort Simpson Airport, we can see that our study period is reasonably representative of long-term conditions for this region – certainly within any observation**

**error (Fig 4).**

Page 14, Lines 29-31: This sentence is a bit confusing. Please revise. **We have edited this sentence in the process of revising the discussion.**

Page 14, Line 31: isoWATFLOOD or WATFLOOD? **isoWATFLOOD. This has been clarified.**

Page 15, Line 4: isoWATFLOOD or WATFLOOD? **Actually, upon re-reading, we feel this pertains to hydrological models in general and have therefore revised our text to be more general.**

Page 15, Line 10: isoWATFLOOD or WATFLOOD? **WATFLOOD. This has been corrected.**

Page 16, Line 13: kpn or KPN43? **Modified to KPN43. Thank you.**

Please check the citations carefully. Pietroniro et al. (1996) and Töyra et al. (1997) are listed in the references section (Page 18, Line 46 and Page 19, Line 34) but appear not in the manuscript itself. **Thank you for noticing this – we have gone through each reference and ensured there is a corresponding citation in-text. We have removed the references you noted were missing citations.**

In general, I liked the style and the coloring of the figures. However, figure 2 and 3 are a bit unclear. It is a real asset to show the uncertainty bounds of the different calibrations. The authors should rethink the presentation of this data, especially the streamflow results (panel b). **You have raised a really interesting perspective here! When we wrote the manuscript and prepared the figures, our interest was in how and where the uncertainty bounds overlapped and were NOT different – but we recognize that to some readers, where they differ is of more interest. Therefore we have darkened and shaded the lines defining each uncertainty envelope so that readers can pick out the uncertainty bands related to each model, and their overlap/differences. (shown on Fig. 5 are the revised panel (b) for Figure 3 Jean Marie and Figure 4 Blackstone, respectively)**

Further more I would suggest indicating periods with snowfall and rainfall, if possible. At this point it would also make sense to combine the two times series (static-rainfall and

static-snowfall) to one static-precipitation input time-series. **Regarding rainfall and snowfall being combined into one time-series, we respectfully disagree since these are two distinct inputs in isoWATFLOOD that can be both used at the same time when there are rain-on-snow events – meaning that both compositions are needed to define the mixed composition of precipitation, where Ptotal represents the sum of snow water equivalent (SWE) and rainfall, used in the model based on: $\delta$P = ($\delta_{rain}$ x RAIN + $\delta_{snow}$ x SWE)/Ptotal. Since both distinct compositions can be used/needed in the same time step, we feel it is important to distinguish the time-series' and show them independently.**

Figure 6: Are here shown the mean or median values (circle symbols)? **We are showing mean values here, and have clarified in figure caption and in methods section.**

Figure 7: Please refer to Table 6 (were the parameters are explained) in the figure caption. **We have added this citation for Table 6.**

The order of the table numbering in the text is sometimes were confused (Page 4, Line 4: Table 1; Page 4, Line 32: Table 4, for example). Please order them correctly. **This has been corrected and tables are now numbered in the order in which they are cited in text.**

In general, I suggest reducing the amount of tables. Table 3 for example is not needed. The applied average correction values (and the range) can be mentioned in the text. Table 5 is also unnecessary. You can mention the values in the text. However, it would be very relevant to explain in more detail how these values were selected. **We have removed Tables 3 and 5 and included this information in the text instead.**

Table 8 is also unnecessary from my point of view. **Given one of the primary goals of this study is to assess the impact of input choice (precipitation isotope product) on the model parameterization, we feel Table 8 contains highly valuable information for tracer-aided modellers tackling the same issues. Therefore, we are inclined to keep it included in our study, but have decided to include it as supplemental information instead of in the manuscript Table S-1).**

**References cited in our response:**

Beven K, and Freer J: Equifinality, data assimilation and uncertainty estimation in mechanistic modelling of complex environmental systems using the GLUE methodology. J Hydrology 249: 11-29, 2001.

Beven, K., and A. Binley: The future of distributed models—Model calibration and uncertainty prediction, Hydrol. Processes, 6(3), 279– 298, 1992.

Connon, R.F., W.L. Quinton, J.R. Craig, J. Hanisch, and O. Sonnentag: The hydrology of interconnected bog complexes in discontinuous permafrost terrain. Hydrol. Process., 29: 3831-3847. 2015. Davison B, Chun K-P, Fortin V, LeConte R, Liu A, Mekonnen M, Stadnyk T, Wheater H. Verification of Gridded Rainfall Products in Central-Western Canada. In preparation.

Holmes, T. L. (2016), Assessing the values of stable water isotopes in hydrologic modeling: A dual-isotope approach. M.Sc. Thesis, University of Manitoba, Winnipeg, 198 pp.

Klaus, J.  McDonnell, J. J., 2013.  Hydrograph separation using stable isotopes: Review and evaluation. Journal of Hydrology, Volume 505, pp. 47-64.

Kling, H.V., and H. Gupta: Decomposition of the mean squared error and NSE performance criteria: Implications for improving hydrological modelling. J. Hydrol., 377: 80-91, 2009.

Lilhare, R.: High-resolution hydrological modelling of the Lower Nelson River Basin, Manitoba, Canada. Poster, ArcticNet, Winnipeg MB, Canada 5-9 December 2016.

Penna, D., M. Ahmad, S. J. Birks, L. Bouchaou, M. Brenčič, S. Butt, L. Holko,G. Jeelani, D. E. Martínez, G. Melikadze, J. B. Shanley, S. A. Sokratov, T. Stadnyk, A. Sugimoto, and P. Vreča (2014), A new method of snowmelt sampling for water stable isotopes, Hydrol. Process., 28, pages 5367–5644, doi: 10.1002/hyp.10273.

Smith A., Delavau C., and Stadnyk., T., 2015.  Identification of geographical influences and flow regime characteristics using regional water isotope surveys in the lower Nelson River, Canada.  Canadian Water Resource Journal, 40 (1):

23-35.

Smith A., Welch C., and Stadnyk., T., 2016. Assessment of a lumped coupled flow-isotope model in data scarce Boreal catchments. Hydrological Processes, 30: 3871-3884.

Stadnyk, T. A., C. Delavau, N. Kouwen, and T. W. D. Edwards (2013), Towards hydrological model calibration and validation: simulation of stable water isotopes using the isoWATFLOOD model. Hydrol. Process., 27, 3791-3810, doi: 10.1002/hyp.9695.

Stadnyk-Falcone, T. A., (2008), Mesoscale Hydrological Model Validation and Verification using Stable Water Isotopes: The isoWATFLOOD Model. Ph.D. Thesis, University of Waterloo, Waterloo, 386 pp., http://hdl.handle.net/10012/3970.

Tetzlaff, D., J. Buttle, S. K. Carey, K. McGuire, H. Laudon, and C. Soulsby (2015), Tracer-based assessment of flow paths, storage and runoff generation in northern catchments: a review. Hydrol. Process., 29, 3475–3490, doi: 10.1002/hyp.10412.

Tolson BA, and Shoemaker CA: Efficient prediction uncertainty approximation in the calibration of environmental simulation models. Water Resources Research, 44:W04411, 2008.

Zak, S. K., and K. J. Beven: Equifinality, sensitivity and predictive uncertainty in the estimation of critical loads, Sci. Total Environ., 236(1–3), 191–214, 1999.
* * *
**JEAN MARIE**

**BLACKSTONE**

'96 / '97 · · · · · KPN43
· · · · · REMOiso
· · · · · static
◇◇◇ $\delta^{18}O_{SNW}$

'97 / '98

'98 / '99

SWE

Precipitation

**Fig. 1.** Revised Fig.5 from manuscript, showing observations of snowpack isotopes

[Figure]

**Fig. 2.** Fig. 3 panel b in log-scale

| Study Period (1997-1999) | Dec-May | June-Nov | TOTAL |
|---|---|---|---|
| Precipitation (TOTAL) (mm) | 350.7 | 956.3 | 1307 |
| *Precipitation (% of total)* | *27%* | *73%* | |
| | | | |
| Snowfall (mm) | 257.4 | 196.6 | 454 |
| *Snowfall (% of total precip)* | *20%* | *15%* | *35%* |
| *Snowfall (% of total snowfall)* | *57%* | *43%* | |
| Rainfall (mm) | 93.3 | 759.7 | 853 |
| *Rainfall (% of total precip)* | *7%* | *58%* | *65%* |
| *Rainfall(% of total rainfall)* | *11%* | *89%* | |

**Fig. 3.** Tabular summary of percent breakdown for seasonal rain and snowfall

| Climate Normal (1981-2010) Fort Simpson A | Dec-May | June-Nov | TOTAL |
|---|---|---|---|
| Precipitation (TOTAL) (mm) | 117.4 | 270.2 | 387.6 |
| *Precipitation (% of total)* | *30%* | *70%* | |
| Snowfall (cm) | 119.9 | 67.2 | 187.1 |
| Snowfall (mm) | 93.6 | 55.5 | 149.1 |
| *Snowfall (% of total precip)* | *24%* | *14%* | *38%* |
| *Snowfall (% of total snowfall)* | *63%* | *37%* | |
| Rainfall (mm) | 23.8 | 214.7 | 238.5 |
| *Rainfall (% of total precip)* | *6%* | *55%* | *62%* |
| *Rainfall(% of total rainfall)* | *10%* | *90%* | |

**Fig. 4.** Tabular summary of climate normal (1981-2010) rain and snowfall

[Figure]

**Fig. 5.** Fig 3 & 4 panel b revised for manuscript

---

## Author Comment (AC4) · 14 Jan 2017

**Abstract.** Tracer-aided hydrological models are becoming increasingly popular tools as they assist with process understanding and source separation; aiding model calibration and diagnosis of model uncertainty (Tetzlaff et al. 2015; Klaus  McDonnell, 2013). Data availability in high-latitude regions, however, proves to be a major challenge associated with this type of application (Tetzlaff et al., 2015). Models require a time series of isotopes in precipitation ($\delta^{18}$O$_{ppt}$) to drive model simulations, and throughout much of the world, and particularly in sparsely populated high-latitude regions, these data are not widely available. Here we investigate the impact that choice of precipitation isotope product ($\delta^{18}$O$_{ppt}$) has on simulated of streamflow, ïĄď18O in streamflow, and model

parameterization in a high-latitude, data sparse, seasonal basin (Fort Simpson, NWT, Canada). We assess three precipitation isotope products of different spatial and temporal resolution (i.e., semi-annual static, seasonal KPN43, and daily REMOiso), and apply them to force the isoWATFLOOD tracer-aided hydrologic model. Although total simulated streamflow was not significantly impacted by choice of $\delta^{18}O_{ppt}$ input, simulated isotopes in streamflow ($\delta^{18}O_{sf}$) and the internal apportionment of water (driven by model parameterization) were impacted. The highest resolution forcing (REMOiso, daily) performed differently than the two lower resolution products (i.e., KPN43 and static), but could not be verified as correct using monthly $\delta^{18}O_{ppt}$ observations. The resolution of a precipitation isotope product impacts model parameterization and seasonal hydrograph separations, where models are most sensitive to large snowmelt and rainfall events when event compositions differ significantly from $\delta^{18}O_{sf}$. Spatial variability in precipitation isotopes was seen and impacts model parameterization, which only distributed tracer-aided hydrological models can represent and respond to. We achieve an understanding of tracer-aided modelling and its applications in high-latitude regions with limited $\delta^{18}O_{ppt}$ observations, and the value these models have in defining model uncertainty. In this study, application of a tracer-aided model was able to identify simulations with improved internal process representation, reinforcing that tracer-aided modelling approaches assist with resolving hydrograph component contributions and work towards diagnosing equifinality.

---

## Author Response (AR1)

Department of Civil Engineering University of Manitoba Winnipeg, MB, Canada Email: tricia.stadnyk@umanitoba.ca Tel: (204) 474-8704 Fax: (204) 474-7516

20 February 2017

Dr. Christine Stumpp Editor Hydrology and Earth System Sciences

**Re: HESS-2016-539**

Dear Dr. Stumpp:

Enclosed please find a fully revised, original manuscript now titled "*Examining the impacts of precipitation isotope products (\delta^{18}O) on distributed tracer-aided hydrological modelling*", which is renamed from the previous title "Examining the impacts of estimated precipitation isotope ( $\delta^{18}O$ ) inputs on distributed tracer-aided hydrological modelling" (reference #HESS-2016-539) by Carly J. Delavau, Tricia A. Stadnyk, and Tegan Holmes. We are respectfully submitting our revised manuscript for your consideration in *Hydrology and Earth System Sciences*.

This manuscript evaluates the impact that different spatial and temporal resolutions of precipitation isotope products ( $\delta^{18}O_{ppt}$ ) have on simulated tracer-aided model output, parameters, and uncertainty. We present three different model calibrations, each derived from a different precipitation isotope product, and statistically assess the behavioural simulations, including: the number of parameter sets retained, differences in parameter distributions, and resulting hydrograph separations and associated parameter uncertainty envelopes. Choice of precipitation isotope product influenced parameter distributions, uncertainty envelopes and resulting hydrograph simulations; but had limited impact on resultant total streamflow simulations. This highlights that tracer-aided models are essential in the diagnosis of equifinality, and in quantifying changes to model output and uncertainty resulting from model input. The higher resolution  $\delta^{18}O_{ppt}$  products were able to reproduce the observed streamflow isotopic variability most reliably, and the highest resolution product (REMOiso) had distinct hydrograph separations relative to the KPN43 and static products. Though this study could not confirm the accuracy of the any one product over another (due to a lack of daily  $\delta^{18}O_{ppt}$  observations), it demonstrated that resolution of tracer-aided model inputs directly impacts model parameterization and resulting hydrograph separations.

We have fully revised the paper to take into consideration the constructive comments from the two referees. Given the manuscript required major revision, we have not provided a line-by-line

list of the changes since line numbers have been altered significantly. Instead, we summarize here the major revisions we have made:

- Rewriting of the abstract
- Broader focus on applications to tracer-aided modelling, rather than study-site specific findings
- Additional methodology section on "statistical treatment of data"
- Rewriting of the discussion to focus more specifically on pertinent results and highlight discrepancies within the modelling (based on reviewer feedback)
- Rewritting of the conclusions that highlight take-home messages from the manuscript and that better connect to our broader objectives.

Of note, in response to reviewer feedback and suggestions we have added a supplement (Table S-1 and Figure S-1), a new methodology section (3.4 Statistical treatment of data), renumbered the figures and tables so they appear consecutively, and enhanced the discussion and conclusions sections of the paper. In response to comments from Anonymous Referee #1, we have inserted some detailed text around our assumption that snowpack and snowfall compositions are equivalent, added the snowpack compositions to Fig 5, and expanded our discussion of the results - particularly as they pertain to streamflow and isotopic simulation errors. In response to Dr. Birkel (Referee #2), we have expended the scope and focus away from the study and onto tracer-aided modelling in general and have included spatial maps of the isotope in precipitation input (Figure S-1). We have not run a configuration of the model over 100K iterations, however, because of time constraints and because we did not feel that parameter identifiability was the overall goal of this study. We will however take this advice and apply it to future studies - which are in fact currently underway. We agree that parameter identifiability is important, however, in this paper, we were more interested in input uncertainty and the impact on the range of parameter uncertainty, which we feel we have addressed. A response document provides the full details of the revisions incorporated in the manuscript.

This manuscript has not been previously published in any language nor is it under consideration for publication by another journal. All authors have carefully read the revised manuscript and have agreed to its submission to Hydrology and Earth System Sciences. River discharge and precipitation time series used in this research were from publically available open sources, and all model results and innovations were developed by the authors using the Fortran programming language and Matlab. Figures were generated using a Grapher package. We are willing to share our  $\delta^{18}O_{ppt}$ models and code with interested researchers upon request (Carly.Delavau@gov.mb.ca).

Please note also that the results presented in this paper originate from the lead author's PhD research under the supervision of the second author. The PhD thesis has been published in the University of Manitoba online repository, and is publically available (http://hdl.handle.net/1993/31946). The results from this paper have not been presented at, nor submitted to, any academic conference; and are not currently nor have not been previously submitted for publication in another journal.

Thank you for your consideration of this contribution to HESS, and to the reviewers for their feedback and edits. We look forward to hearing from you.

Sincerely yours,

Dr. Tricia Stadnyk, P.Eng.

Corresponding Author

**RESPONSE TO THE REFEREES' COMMENTS**

We sincerely thank both referees for their thorough reviews and most constructive comments on our manuscript (Reference # HESS-2016-539). We fully recognize and appreciate the reviewers' efforts in providing these informative reports on our research and their insights have led to an improved interpretation of our results. We have therefore taken into full consideration all of these comments and have prepared responses to these as well as information on how the paper was revised following the referees' suggestions. Our responses to reviewers are provided below **in bold** following the individual comments requiring action from both reviewers, followed by a marked up version of the manuscript (changes highlighted in yellow).

Referee #1:

Specific comments:

The authors should rethink the use of the word "estimated" in the title as well as throughout the whole manuscript. It suggests that the input data was generated specifically for the presented study. It should be clear that (2 of 3) available precipitation isotope product were used to the study. Which is actually an asset for the study and with respect to future studies in other basins.

We agree completely, and this was also suggested by the second reviewer too. We have changed the title to "*Examining the impacts of precipitation isotope products* ( $\delta^{18}O$ ) on distributed tracer-aided hydrological modelling" and revised the use of the word 'estimated' (with respect to  $\delta^{18}O_{ppt}$  inputs) throughout the manuscript to "precipitation isotope products", as appropriate.

The first sentence of the abstract is "...increasingly popular tools as they have documented utility in constraining model parameter space during calibration, reducing model uncertainty, and

assisting with selection of appropriate model structures.". However, there is no evidence for that statement. Please include additional information to the introduction section or revise the first sentence of the abstract.

We have subsequently revised the abstract significantly, and agree that it has yet to be proven that the parameter space is constrained by such tools. We have rephrased this sentence as: "*Tracer-aided hydrological models are becoming increasingly popular tools as they assist with process understanding and source separation; which facilitates model calibration and diagnosis of model uncertainty (Tetzlaff et al. 2015; Klaus & McDonnell, 2013)*".

The authors highlight the importance of snowmelt in the study region. The stable isotope signature of the snow pack and its melt water is a very challenging topic. Please handle this point very carefully in your publication. On page 5, Line 17 for example you mention that the default method for oxygen-18 input is annual average rainfall and snowfall. In your static approach, however, you used average measurements of rainfall and snowpack from the GEWEX campaign. Please provide the values of snow pack stable isotope signature in figure 5 by the way. Especially during the ablation season the isotopic evolution of the snowpack progresses due to percolating rain water and fractionation caused by processes like melting and sublimation (Zhou et al., 2008; Unnikrishna et al., 2002; Dietermann and Weiler, 2013; Lee et al., 2010). This leads to an increase of heavy isotopes in melt water throughout the freshet period (Taylor et al., 2001, 2002; Unnikrishna et al., 2002). Which is correctly represented by the shown model results. Taylor et al. (2001 and 2002) point out that for hydrological applications (in their case isotope based hydrograph separation) a correct representation of the snow pack melt water is absolutely crucial.

Thank you for your insight, and we couldn't agree more that the isotopic signature of a snowpack and its evolution in snow melt are very challenging processes. We have since revised Figure 5 and added the snowpack data and also included a cautionary note to readers highlighting there is uncertainty surrounding these measurements. For the modelling, as a static input our model would preferably use average annual inputs of rainfall and snowfall. Rainfall and snowpack values were obtained from the GEWEX campaign. There was no data on snowfall composition available – only snowpack compositions - therefore we assume (as model input) that the average annual composition of snowfall is approximately equal to that of the snowpack. We have clarified our assumption in the manuscript.

REMOiso is a distributed dataset and the precipitation amounts are also available spatially distributed over the study area. Why was the precipitation amount weighting only conducted at one location and not spatially distributed?

The only precipitation amount-weighting for REMOiso was done to determine the bias correction at Snare Rapids. There was no need to do this spatially for this purpose as we are comparing CNIP observations (at a point) directly to REMOiso output at a single location corresponding to the location of the CNIP observation station. We averaged the four 6-hourly REMOiso values (at each grid) to arrive at daily compositions that were read into the model as input on a per-grid basis (i.e., no amount-weighting involved – same as for the static and KPN inputs). Based on some (unpublished) analyses we did for a study of

the Mackenzie River Basin (i.e., using the same REMOiso model output), we don't trust the quality of sub-daily REMO precipitation to the point where we would use (sub-daily) precipitation to amount weight REMOiso  $\delta^{18}O_{ppt}$ . If we decided to amount weight, we couldn't use actual observations to amount weight 6-hourly to daily as we only have daily precipitation from Fort Simpson Airport and at various grid locations from the ANUSPLIN product.

The authors mention that "several changes and improvements" (Page 7, Line 16) were carried out in the model version used for the study. In the following only one modification (proportion of bog an fen split) is mentioned. Are there any other modifications? If so, please mention them here.

This was poorly worded on our part. What we meant to say was that the model (isoWATFLOOD) has undergone "several changes and improvements" since it was last published in a study back in 2013 (Stadnyk et al., 2013). These changes and improvements were independent of the current study, and all toward continual improvement of internal dynamics and the model output. We have revised the wording in our manuscript to clarify: "*The isoWATFLOOD model used in this study is based on a previous version used by Stadnyk et al. (2013). The current model, however, uses an updated version of isoWATFLOOD code and the watershed set-up incorporates various model improvements made since 2013, independent of this study.*"

The first two paragraphs of section 4 (Results and discussion) should definitively be revised. There is a lot of content that can be mentioned later in the conclusions section (the last sentence on Line 12-14 for example).

We have significantly revised the results & discussion using the guidance of your questions below to help highlight specific findings related to our key objectives and take-home messages. We have also moved the sentence you reference above to the conclusions.

In section 4.2 (Modelling streamflow) please explain the model results as well as the observed streamflow in much more detail. The three different inputs (and three different calibrations) provide very similar results for the simulated streamflow (Page 15, Lines 5-8). Those results should be discussed in more detail.

Thank you for your suggestions, we have revised the discussion to include more specific, indepth discussion of the simulated streamflow resulting from the three types of precipitation isotope product. And yes, all three precipitation isotope products (three different calibrations) result in almost exactly the same streamflow simulation (i.e., statistically the same according to the Kendall's tau test applied in the paper).

Is there really no discharge in winter (Figure 2 and 3)?

We assume you are referring to Figures 3 & 4 (not 2). And no, observed streamflow does not go zero, but rather becomes very small relative to peak flows: minimum in Jean-Marie from 1997-1999 of 0.194 m3/s, or 0.5% of the maximum streamflow, 35 m3/s during this same period; and a minimum of 0.043 m3/s in Blackstone relative to a maximum flow of 109 m3/s, so less than 0.04% of the peak flow. We have added the average ice-on flows over the study period to the study site/background section for clarity. Ice-on winter low flows in

high latitude basins such as this commonly reduce significantly and become near zero due to the long, sustained period frozen ground/soils, lack of mid-winter thaw/melt periods, and accumulation of solid precipitation.

We considered providing panel b on Fig 3 & 4 in log-scale to emphasize that there are in fact low-flow values; but this greatly diminished peak flow analysis and peak flow uncertainty, which was a key point in our study. We have included those log-scale figures here for your assessment (not included in revised manuscript):

What are the influences of groundwater on the hydrology of the region? The same holds for section 4.3. Explain the results in more details.

Given the region resides within the discontinuous to semi-permafrost region of Canada, the influence of sub-surface contributions to runoff would be sporadic and is difficult to define (as several studies in the region have shown, Connon et al., 2015). The model we use in this study (isoWATFLOOD) has the capability to raise/lower wetland water table levels, connecting and/or disconnecting with channel runoff, which is a reasonable analogy to this complex interaction.

There is especially the time of the spring freshet that needs much more carefully discussed. We have incorporated an analysis of the results during spring freshet into our discussion.

The model results show a sharp drop of streamflow stable isotope signature, while the observed values are getting more and more enriched at that time. This completely opposed development may be related that the contribution of snowmelt water to total streamflow during that time is too high (or the signature of the snow melt signal is wrong, please remind here my suggestions above) and the contribution of baseflow too low.

We assume you are referring to the freshet period in 1998. Note that we did not have continuously observed isotopes in streamflow during the peak freshet (i.e., high flow sampling is not always feasible), and as a result there are some missing observations during this time of year (mostly in 1999), despite this being our most frequent period of sampling overall (relative to other seasons). Moreover, as we've explained the model assumes snowfall composition to be equal to snowpack composition, and then can apply a constant offset or fractionation from snowpack composition/accumulation to snowmelt. In this study, that offset was set =0 given the lack of snowpack to snowmelt observations from which to calibrate to. Therefore, it is most likely that, in this year, the assumed fractionation from pack to melt water was wrong and not well defined. Again – without observed data to compare to, it is impossible for us to adjust this factor to improve results;

**however, adding a snowmelt dynamics module to the model would be a great asset, one which has been recognized by our group and that we are working toward. We have added some text in the revised manuscript to discuss this discrepancy.**

The contributions of baseflow (groundwater) to total streamflow during the post-freshet are especially for Jean Marie River much lower in the present model study compared to the results of St Amour et al. (2005). Furthermore, please provide the stable isotope signature of groundwater. And compare those observed values with the values generated by the models in the groundwater routine after the spin-up period.

As Stadnyk et al. (2005) and Stadnyk-Falcone (2008) pointed out, contributions of "groundwater" from the model (isoWATFLOOD) cannot be directly compared to those derived by St. Amour et al. (2005) owing to the definition of what groundwater is considered in the two modelling methodologies. In St. Amour et al (2005), a mixing model is used that separates old and new water contributions over time – which means that groundwater is defined as old water, or that is water that is existing pre-event. Whereas using WATFLOOD or isoWATFLOOD to perform hydrograph separation in the same region, lower contributions of groundwater are derived by the (iso)WATFLOOD model since the model separates soil water (upper zone storage) from baseflow or groundwater (lower zone storage) and wetland storage -- all of which would constitute 'old' (pre-event) water using traditional two-component mixing models. We have added text in the revised manuscript to describe this.

Please check the manuscript for repetitive information. The sentence on Page 13, 34+35 for example appears almost identical on the next page again (Page 14, Lines 20-22). This would be an excellent take-home sentence for the conclusions section by the way.

Thank you for pointing this out. We have re-read the manuscript and removed any apparent redundancies, particularly the ones you have pointed out to us. We have moved the sentence you highlighted to the conclusions section.

In general, I am missing some distinct conclusions in the conclusions section of the submitted manuscript. There are a lot of recommendations and speculations but no clear take-home messages.

Agreed. In re-reading the manuscript, we too realized that we can write better conclusions that highlight the take-home messages this manuscript presents. Also elaborated on in the conclusions now is the take-home message that precipitation isotope products of higher resolution (e.g., REMOiso, daily resolution) better capture event-specific compositions that, when significantly different from  $\delta^{18}O_{SF}$ , tend to cause significant deviations from seasonal and semi-annual (i.e., static) inputs. Though we cannot verify the correctness of the higher resolution product (REMOiso) in this study due to monthly observed precipitation, it is clear that temporal resolution plays a significant role in model parameterization and resulting hydrograph separations. We have also added a separate "*Future Directions*" section (based on Reviewer #2 feedback) that is comprised of the future work discussion from our original conclusions.

Technical notes:

Page 1, Line 18: "...to capture both the variability and seasonality". There it would be better to write "spatial variability and seasonality" or "spatial and temporal variability", since the seasonality is also a variability (temporal).

**We have made this correction.**

Page 1, Line 31: (e.g. Beven and Binley, 1992; Kirchner....)
Correction made.
Page 3, Line 22 and Line 29: Please provide size and elevation characteristics of the basins here.
We have added this information.

Page 3, Line 27: "...is selected based ON data availability." Correction made.

Page 4, Line 20: The study region is not a high elevation region. Please mention correctly why the approach is suitable for the study region.

From another project our research group is working on, a detailed analysis of ANUSPLIN's suitability for high-latitude, Boreal regions (i.e., specifically the Nelson River) was done by a PhD student (Rajtantra Lilhare) and presented recently in a poster at ArcticNet (Lilhare, 2016). In this study, both the seasonality and amount of precipitation from ANUSPLIN were found to match well (r>=0.98) with nearby observations (3 for precipitation, 6 for temperature; all within the Nelson River watershed) from Provincial and Environment Canada meteorological station observations (shown here, but not included in our paper).

**ANUSPLIN Temperature**

---

## Author Response (AR2)

Department of Civil Engineering
University of Manitoba
Winnipeg, MB, Canada
Email: tricia.stadnyk@umanitoba.ca
Tel: (204) 474-8704
Fax: (204) 474-7516

31 March 2017

Dr. Christine Stumpp
Editor
Hydrology and Earth System Sciences

**Re: HESS-2016-539**

Dear Dr. Stumpp:

Enclosed please find our revised manuscript titled **"*Examining the impacts of precipitation isotope products ($\delta^{18}O$) on distributed tracer-aided hydrological modelling*"** (reference #HESS-2016-539) by Carly J. Delavau, Tricia A. Stadnyk, and Tegan Holmes. We are respectfully submitting our minor revisions in response to reviewer #1 for your consideration to be published in *Hydrology and Earth System Sciences*.

This manuscript evaluates the impact that different spatial and temporal resolutions of precipitation isotope products ($\delta^{18}O_{ppt}$) have on simulated tracer-aided model output, parameters, and uncertainty. We present three different model calibrations, each derived from a different precipitation isotope product, and statistically assess the behavioural simulations, including: the number of parameter sets retained, differences in parameter distributions, and resulting hydrograph separations and associated parameter uncertainty envelopes. Choice of precipitation isotope product influenced parameter distributions, uncertainty envelopes and resulting hydrograph simulations; but had limited impact on resultant total streamflow simulations. This highlights that tracer-aided models are essential in the diagnosis of equifinality, and in quantifying changes to model output and uncertainty resulting from model input. The higher resolution $\delta^{18}O_{ppt}$ products were able to reproduce the observed streamflow isotopic variability most reliably, and the highest resolution product (REMOiso) had distinct hydrograph separations relative to the KPN43 and static products. Though this study could not confirm the accuracy of the any one product over another (due to a lack of daily $\delta^{18}O_{ppt}$ observations), it demonstrated that resolution of tracer-aided model inputs directly impacts model parameterization and resulting hydrograph separations.

We have made the minor revisions requested by reviewer #1 and addressed their concerns, and have summarized these changes (suggested by reviewer #1) as follows (our responses are indicated in **bold font**):

- *Page 11 (section 4.2): The authors explain "differences in hydrograph characteristics" with "variations in basin physiography, storage mechanisms and land cover composition". The differences especially for the peak flows are considerably for the two basins having almost identical catchment size. Please explain the differences in runoff volume in more detail. Discharge in millimeters (standardized to catchment area) would be used to allow a direct comparison of the two basins.*
  **In an attempt to clarify, we have added a normalized discharge comparison to the manuscript, and added some text stating that the basins have a very similar drainage area and runoff volume, but different runoff responses (which we attribute to basin physiography such as basin slope and landcover).**

- *The streamflow simulation shows very similar results using the three different isotope products. This is not surprisingly if the same precipitation input (with different stable isotope signature) was used for the different simulations. Please clarify this issue.*
  **We understand the reviewers point and would like to point out that the remainder of the manuscript focuses on exactly this: the issue of equifinality. Our take home point is that based on streamflow alone, one may assume the two basins are equal in runoff response. However, the use of isotopes in the modelling shows that this is not the case, nor should one expect that to be the case (and that the parameters internally in the model change to reflect these differences in landcover driving runoff response). More specifically, we have a paragraph on page 12 basically saying that the three precipitation isotope products generate statistically similar streamflow simulations. Then we go on to show the differences in parameterization and apportionment of water.**

- *Page 13, Line 26: Please mention that this was a bulk sample of the event.*
  **Done.**

- *Page 15, Line 23: Stadnyk et al. (2013) instead of Stadnyk et al. (2005) since you are referring to isoWATFLOOD here.*
  **Done. Thank you for noticing this!**

- *Page 16, Line 20: KPN43*
  **Done.**

This manuscript has not been previously published in any language nor is it under consideration for publication by another journal. All authors have carefully read the revised manuscript and have agreed to its submission to *Hydrology and Earth System Sciences*. River discharge and precipitation time series used in this research were from publically available open sources, and all model results and innovations were developed by the authors using the Fortran programming language and Matlab. Figures were generated using a Grapher package. We are willing to share our $\delta^{18}O_{ppt}$ models and code with interested researchers upon request (Carly.Delavau@gov.mb.ca).

Please note also that the results presented in this paper originate from the lead author's PhD research under the supervision of the second author. The PhD thesis has been published in the University of Manitoba online repository, and is publically available (http://hdl.handle.net/1993/31946). The results from this paper have not been presented at, nor submitted to, any academic conference; and are not currently nor have not been previously submitted for publication in another journal.

Thank you for your consideration of this contribution to HESS, and to the reviewers for their feedback and edits. We look forward to hearing from you.

Sincerely yours,

Dr. Tricia Stadnyk, P.Eng.

Corresponding Author

[revised manuscript text omitted]
 4. It is worth noting that both basins have similar drainage areas and received comparable precipitation inputs over the study period, which would naturally result in similar streamflow responses. Comparing normalized (by drainage area) observed discharge over the study period for the basins reveals the Blackstone sub-basin generates nearly twice as much runoff as the Jean Marie sub-basin, with normalized discharges of 0.56 mm/km$^2$ and 0.31 mm/km$^2$, respectively. Therefore, differences in hydrograph characteristics (i.e., peak flows, attenuation, etc.) between Jean Marie and Blackstone result from variations in basin physiography, storage mechanisms, and land cover composition; specifically large differences in average basin slope and surface water and wetland dynamics (St Amour et al., 2005). Namely, the higher energy environment of Blackstone River promotes a quicker runoff response; and the flatter, more surface water dominated Jean-Marie basin yields a damped runoff response, on average.

[revised manuscript text omitted]